## METHOD

# MAAT: a new nonparametric Bayesian framework for incorporating multiple functional annotations in transcriptome-wide association studies

Han Wang[1], Xiang Li[2], Teng Li[3], Zhe Li[4], Pak Chung Sham[5,6] and Yan Dora Zhang[2*]

*Correspondence:
doraz@hku.hk

[1] College of Science, China Agricultural University, Beijing, China
[2] Department of Statistics and Actuarial Science, School of Computing and Data Science, The University of Hong Kong, Hong Kong SAR, China
[3] Department of Medical Oncology, National Cancer Center/National Clinical Research Center for Cancer/Cancer Hospital, Chinese Academy of Medical Sciences and Peking Union Medical College, Beijing, China
[4] 4+4 Medical Doctor Program, Chinese Academy of Medical Sciences and Peking Union Medical College, Beijing, China
[5] Department of Psychiatry, Li Ka Shing Faculty of Medicine, The University of Hong Kong, Hong Kong SAR, China
[6] Centre for PanorOmic Sciences, Li Ka Shing Faculty of Medicine, The University of Hong Kong, Hong Kong SAR, China

## Abstract

Transcriptome-wide association study (TWAS) has emerged as a powerful tool for translating the myriad variations identified by genome-wide association studies (GWAS) into regulated genes in the post-GWAS era. While integrating annotation information has been shown to enhance power, current annotation-assisted TWAS tools predominantly focus on epigenomic annotations. When including more annotations, the assumption of a positive correlation between annotation scores and SNPs' effect sizes, as adopted by current methods, often falls short. Here, we propose MAAT expanding the horizons of existing TWAS studies, generating a new model incorporating multiple annotations into TWAS and a new metric indicating the most important annotation.

**Keywords:** Transcriptome-wide association studies (TWAS), Functional annotation, Product partition model with covariates (PPMx), Psychiatric traits

## Background

The last two decades have witnessed the prosperity of genome-wide association studies (GWAS) in investigating the genetic underpinnings of various diseases [1, 2]. Although GWAS have discovered tens of thousands of genetic variants associated with a myriad of complex traits and diseases [3], it is hard to ascertain the causal genes mediating variant effects on the trait. Motivated by the need for prioritizing candidate genes at GWAS loci, transcriptome-wide association studies (TWAS) have been proposed to pinpoint genes affecting human diseases due to the transcriptional activity [4]. A typical TWAS first trains per-gene expression imputation models utilizing genotype data of cis-SNPs from the reference panel (e.g., the Genotype-Tissue Expression (GTEx) project [5]). Subsequently, it conducts a gene-based association analysis between genetically regulated expression (GReX) and phenotypic data from the GWAS. Many algorithms have been

proposed to implement TWAS with individual-level or summary-level GWAS data, such as PrediXcan [6], Fusion [7], TIGAR [8], S-PrediXcan [9] , T-GEN [10], and UTMOST [11].

Besides integrating information on gene expression regulation, which facilitates TWAS to gain more power in detecting associations compared with SNP-based GWAS, it is of substantial interest to incorporate functional annotations to further enhance the TWAS power. Functional annotations such as contributions to protein function, conservation, and mappability scores have been successfully utilized in GWAS and fine-mapping studies to prioritize important variants [12–14]. For TWAS, epigenomic data are the most commonly used annotations. Under the assumption that SNPs with active epigenomic annotations are more likely to be functionally important in gene expression regulation, recent studies have gained significant improvement in detecting gene-trait associations [10, 15]. Besides epigenomic perspective, 3D genomic data has also been incorporated into the imputation step by recent studies [16]. However, the impact that SNPs impose on gene expression cannot only be reflected by the epigenomic or 3D genomic aspect. Many cross-tissue expression quantitative trait locus (eQTLs) are enriched in transcription factor (TF) binding sites [17]. Conserved DNA regions, although only accounts for a small fraction of human genome [18], can explain a considerable fraction of heritability for complex diseases [19]. Hence, there is a pressing need to aggregate the all-round effects that different annotations provided on genomic function when performing TWAS.

In this paper, we propose MAAT (multiple annotation-assisted TWAS), a new framework that incorporates SNPs' multifaceted annotation information to facilitate more gene-trait association discoveries. We include a diverse array of highly variable annotations in MAAT from Functional Annotation of Variants Online Resources (FAVOR) [14, 20], involving multiple integrative-aspect annotations such as epigenetics principal component (PC), conservation PC, TF PC [14], and FATHMM-XF [21]. A product partition model with covariates (PPMx) [22] is proposed to allocate similar effect sizes for cis-SNPs with analogous annotation profiles. The flexible non-parametric prior in PPMx breaks through the prevalent assumption that higher annotation scores correspond to greater cis-SNP effect size [10, 15, 16]. When annotation data extends far beyond epigenomic or 3D genomic information, especially with the inclusion of integrative annotation scores, the linear relationship adopted by existing methods can hardly capture the complex architecture of influences that different annotations impose on cis-SNPs. Notably, in contrast to conventional TWAS culminating in the identification of significant genes, we take a further stride. Leveraging an angle-type metric, we allocate the paramount annotation for every significant gene-trait association, which signifies the pivotal annotation in a gene's influence on a trait. To the best of our knowledge, there is no method that assigns annotation label in the gene-trait association to date. This novel angle-based approach offers an enhanced avenue for dissecting the etiology of intricate diseases.

We conduct a spectrum of simulation studies to demonstrate that MAAT achieves greater imputation $R^2$ and increased TWAS power to implicate associations while maintaining low type I error rates in different settings. We subsequently apply MAAT to eight GWAS summary datasets pertaining to eight psychiatric traits, which leads

to the discovery of more gene-trait associations compared to current state-of-the-art methods. Through extensive dataset exploration, we show that many of the assigned annotation can be legitimately verified.

## Results

### Method overview

MAAT is a versatile framework designed to incorporate a broad set of functional annotations into TWAS. These annotations cover various aspect of variant function-ality ranging from epigenetic function, TF, mappability to evolutionary conservation, etc. For computational simplicity, we employ the annotation principal component (aPC) and other integrative annotation scores in FAVOR, allowing for simultaneous dimension reduction and information preservation (Methods section).

In the realm of traditional TWAS, two steps are indispensable—the per-gene expression imputation and the association testing step. In MAAT, we keep the second step intact, while refining the first step and marking a stride beyond the second step. Specifically, as shown in Fig. 1, MAAT distinguishes itself from other methods in two pivotal aspects: (i) By employing a non-parametric PPMx prior, we seamlessly inte-grate seven integrative annotation scores into the gene expression imputation step. (ii) Within the TWAS framework, we delve into the question of which annotation exerts the most influence when a gene impacts a disease.

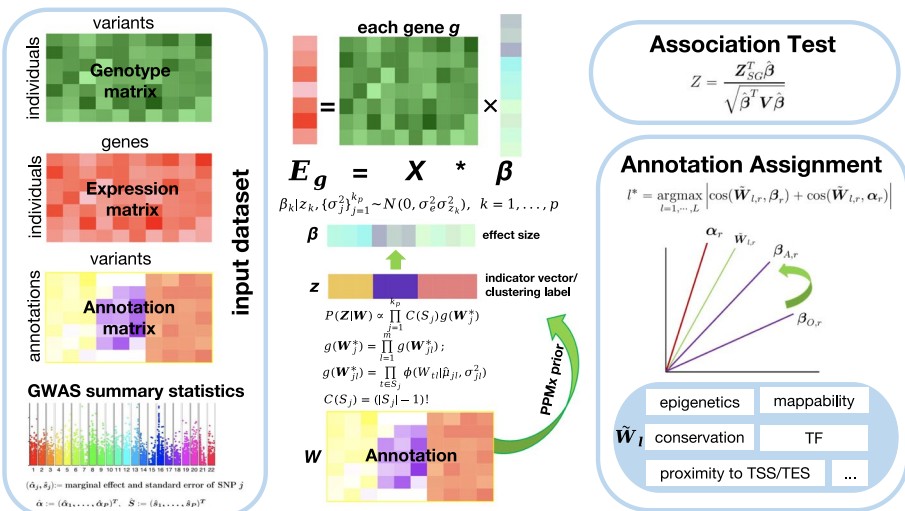

**Fig. 1** Workflow of MAAT. The input files of MAAT are genotype matrix, gene expression matrix, annotation matrix, and GWAS summary statistics. For each gene *g*, MAAT adopts PPMx, a non-parametric Bayesian prior to incorporate multiple annotation information into the imputation model. After imputation, MAAT utilizes GWAS-summary statistics and the effect size obtained in the imputation step to perform gene-trait association analysis. Furthermore, MAAT takes an angle-based metric to assign an annotation for each gene-trait association, which facilitates the understanding of the mechanisms through which genes influence diseases. In the sub-figure for annotation assignment, $\boldsymbol{\alpha}_r$ denotes the rotated effect size that SNPs impose on phenotype, $\tilde{\boldsymbol{W}}_{l,r}$ is the rotated *l*-th annotation, $\boldsymbol{\beta}_{A,r}$ and $\boldsymbol{\beta}_{O,r}$ are the rotated effect size that SNPs impose on gene expression estimated by MAAT and other without-annotation method, respectively. The annotation information in MAAT includes epigenetics, conservation, mappability, transcription factor (TF), proximity to TSS/TES, proximity to coding, and FATHMM-XF

Specifically, we assume $E_g \in \mathbb{R}^n$ is the expression profiles of gene $g$, $X \in \mathbb{R}^{n \times p}$ is a gene-specific genotype matrix, containing the cis-SNP genotype information of gene $g$, $\rho = \{S_1, S_2, \cdots, S_{k_p}\}$ is a partition of the $p$ cis-SNPs into $k_p$ clusters. Each $\rho$ corresponds to a latent vector $Z = (z_1, \cdots, z_p)$, with $z_k = j$ if the $k$-th cis-SNP belongs to cluster $j$ under partition $\rho$. Let $W = (W_1^\top, \cdots, W_p^\top)$ be the annotation matrix, where $W_k = (W_{k1}, \cdots, W_{km})$ represents the $m$ annotation scores assigned to cis-SNP $k$. Each $\rho$ also corresponds to a partition on $W$, we let $W_j^* = \{W_k : k \in S_j\} \in \mathbb{R}^{m \times |S_j|}$ denote the cluster $j$-specific annotation information. Given partition $\rho$, we assume cis-SNPs within the same cluster possess similar effect sizes:

$$E_g = X\beta + \epsilon = X\tilde{\beta} + u + \epsilon;$$
$$\tilde{\beta}_k | z_k = j, \sigma_e^2, \sigma_j^2 \sim N(0, \sigma_e^2 \sigma_j^2), \text{ for } k = 1, \cdots, p;$$

where $u$ is a random effect term to simplify computation (Methods section). We integrate the annotation information $W$ into the imputation step by imposing a PPMx prior on $Z$:

$$P(Z|W) \propto \prod_{j=1}^{k_p} g(W_j^*) C(S_j).$$

Where $g(W_j^*)$ is a similarity function and $C(S_j) = (|S_j| - 1)!$ is a cohesion function (Methods section). When $Z$ corresponds to a rational cis-SNP partition, in the sense that SNPs with similar annotation information are grouped together, the PPMx prior assigns a higher prior probability to such $Z$. This encourages SNPs with similar annotation profiles to share similar effect sizes.

As illustrated in Additional file 1: Fig. S1, for every gene, we calculated the correlation between each pair of annotations, yielding 21 correlation scores for each gene. Out of the 10,940 genes analyzed, 6692 exhibited one or more negative correlation scores among the 21 annotation pairs, constituting over 60% of all genes. This underscores that the existing annotation-assisted TWAS methods, which presume a positive correlation between all types of annotation and cis-SNPs' effect sizes, are not universally applicable. This discrepancy inevitably leads to an accuracy loss in the imputation model and a reduction in TWAS power. In contrast, our proposed PPMx model does not confine itself to the assumption of a linear relationship between the annotation profile and cis-SNPs' effect size. Instead, it posits that cis-SNPs with similar annotation profiles possess similar effect sizes on gene expression. PPMx assigns a prior probability to each potential clustering scheme for cis-SNPs. If a clustering scheme precisely groups similar cis-SNPs together, PPMx grants it a higher prior probability, seamlessly integrating the affluent annotation information into the imputation step (Methods section). In the second association step, a burden $Z$-score, akin to that in FUSION [7], is calculated when only summary-level GWAS data are available.

Finally, benefit from the inclusion of annotation data, one important capability of MAAT lies in its ability to discern which annotation exerts the most substantial influence when genes impact traits within the TWAS framework. Through calculations,

we derived that the power of TWAS can be approximated by the angular distance between two rotated effect size vectors—one representing the effect size of cis-SNPs on phenotypes, and the other representing the effect size of cis-SNPs on gene expression. For each annotation, MAAT assesses the contribution by computing the angular distance between the rotated annotation profile and the two rotated effect size vectors. If the two angular distances are small, MAAT designates this annotation as an important annotation [23, 24] (Methods section and Additional file 1).

### Simulation

To illustrate the advantage of MAAT compared with existing TWAS methods, we performed simulation studies with varying causal SNP proportion $p_{cs}$ (0.01, 0.05, 0.1, and 0.2), expression heritability $h_e^2$ (0.1, 0.2, and 0.5), and phenotype heritability $h_p^2$ (0.1, 0.25, and 0.5). The genotyped and imputed genetic data for the 1000 cis-SNPs (with minor allele frequency (MAF) $\geq 5\%$ and Hardy-Weinberg $p$ value $\leq 1 \times 10^{-7}$) of the randomly selected gene TPTE were used as the genotype information for simulation. For each SNP, five different annotation scores were simulated to be allocated to it, with two informative annotations and three non-informative ones (Methods section). For each ($p_{cs}$, $h_e^2$) scenario, we repeated the simulations 50 times. At each time of a specific ($p_{cs}$, $h_e^2$) scenario, where the expression level and effect size vector are fixed, we further simulated the phenotype 100 times with respect to each $h_p^2$ value to evaluate the performance of power. With respect to type I error, we simulated the quantitative phenotype directly from the standard normal distribution to ensure that the phenotype is independent of the genotype data. Similar to the procedure for evaluating power, for each replication in each ($p_{cs}$, $h_e^2$) scenario, the phenotype was simulated 100,000 times (Methods section). Four widely used TWAS methods (PrediXcan, TIGAR, T-GEN, and EpiXcan) were compared with MAAT with respect to the performance of imputation $R^2$, TWAS power, and type I error rate.

Regarding imputation $R^2$, besides different combinations of $p_{cs}$ and $h_e^2$, we also evaluated the $R^2$ at different sparsity thresholds of the effect size vector $\beta$ after data postprocessing (Additional file 1). As shown in Fig. 2, when there are few causal SNPs and $h_e^2$ is small, all five methods achieve comparable performance; while for other scenarios, MAAT outperforms other methods evidently. For instance, when $p_{cs} = 0.1$, $h_e^2 = 0.5$,

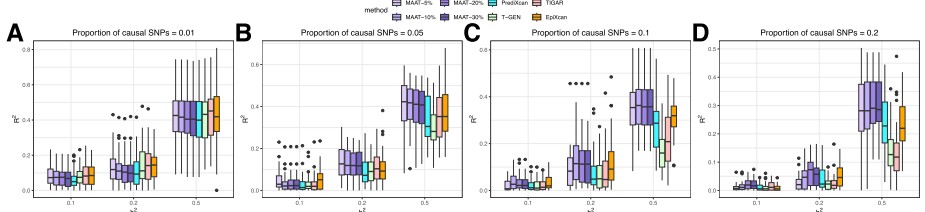

**Fig. 2** Performance of imputation $\boldsymbol{R^2}$ in simulation studies. With regard to the performance of imputation $R^2$, we compare MAAT under different causal SNP proportion thresholds (5%, 10%, 20%, and 30%) with PrediXcan, T-GEN, TIGAR, and EpiXcan under different settings. The true proportion of causal SNPs $p_{cs}$ is set to 0.01, 0.05, 0.1, and 0.2, respectively, which corresponds to **A**, **B**, **C**, and **D** accordingly. The *x*-axis refers to the expression heritability $h_e^2$, which is set to 0.1, 0.2, and 0.5, respectively. For each scenario, each method is replicated for 50 times, the boxplots depict the distribution of the imputation $R^2$ for each method and each scenario

the average imputation $R^2$ of 50 replications ranges from 0.339 (with 5% sparsity threshold) to 0.361 (with 10% sparsity threshold) by MAAT. While the average imputation $R^2$ for the other four methods ranges from 0.168 (T-GEN) to 0.311 (EpiXcan). When the sparsity of $\beta$ is post-processed into different levels, the imputation $R^2$ does not vary too much for fixed $p_{cs}$ and $h_e^2$, especially for scenarios with lower $p_{cs}$. This indicates that even if we do not have a good knowledge of the sparsity level for $\beta$, overestimating the number of causal SNPs does not adversely affect the $R^2$ performance, which demonstrates that MAAT indeed assigns larger effect sizes to true causal SNPs. When $p_{cs}$ is relatively large, underestimation of causal SNPs may induce a decay in $R^2$ slightly. But when $h_e^2$ is relatively high (0.2 or 0.3), even if we underestimate the number of causal SNPs, MAAT exhibits superior imputation $R^2$ performance compared to other methods.

For the calculation of power and type I error, we adpoted aggregated Cauchy association test (ACAT) [25, 26] to combine results evaluated at different sparsity levels of $\beta$ (Methods section). As shown in Fig. 3A–D, the performance of power exhibits a similar pattern to that of $R^2$, a larger $R^2$ induces a larger power. When $h_p^2 = 0.25$, except for scenarios with small proportion of causal SNPs, MAAT achieves the best performance of power for all other scenarios. It can be also observed that, even if $R^2$ is reduced when $p_{cs}$ is underestimated compared to the true $p_{cs}$, MAAT demonstrates a satisfactory level of power by integrating the results at different sparsity levels of $\beta$ through ACAT. For example, when $h_e^2 = 0.2$ and $p_{cs} = 0.2$, the imputation $R^2$ of MAAT at 5% sparsity level is 0.028, which is smaller than PrediXcan, T-GEN, TIGAR, and EpiXcan. But when the results across different sparsity levels of $\beta$ are aggregated, thanks to the good performance of MAAT at other sparsity levels, the resulting power of MAAT is larger than the other four methods. When we set $h_p^2 = 0.1$ or 0.5, a similar trend in power variation can be observed. (Additional file 1: Figs. S2–S3). It is also evident that when the

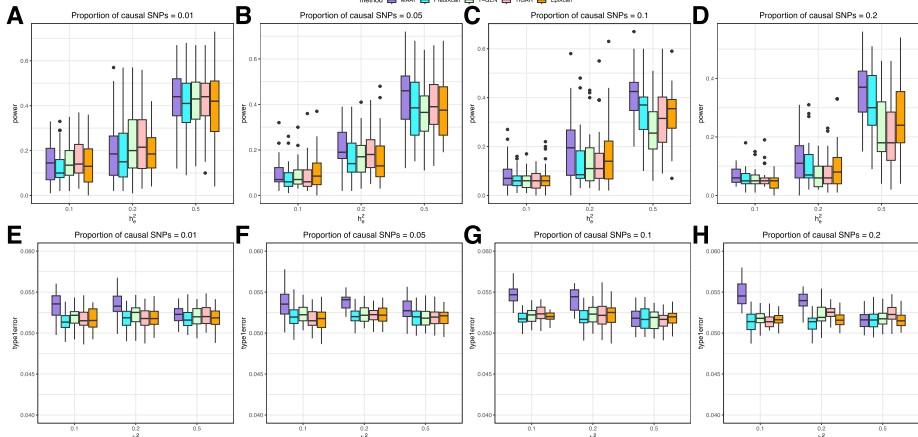

**Fig. 3** Performance of power and type I error in simulation studies. We compare MAAT with PrediXcan, T-GEN, TIGAR, and EpiXcan under different settings to evaluate the performance of association test power (**A–D**) and type I error (**E–H**). The true proportion of causal SNPs $p_{cs}$ is set to 0.01, 0.05, 0.1, and 0.2, respectively. The *x*-axis refers to the expression heritability $h_e^2$, which is set to 0.1, 0.2, and 0.5, respectively. The phenotypic heritability $h_p^2$ is set to 0.25 when calculating the power. At each replication of a ($p_{cs}$, $h_e^2$) setting, the phenotype is replicated for 100 times for power analysis and 100,000 times for type I error analysis; the power and type I error are calculated as the proportion of *p* values reaching the significant level among the 100 replications and 100,000 replications, respectively. The boxplots depict the distribution of power (**A–D**) and type I error (**E–H**) for each scenario and each method

phenotype heritability increases, there will be a notable improvement in TWAS power. With respect to type I error, all five methods can control the false discovery rate (FDR) within 6%. When $h_e^2$ and $p_{cs}$ are relatively small, MAAT tends to have a FDR approximately 0.3% higher than other methods. However, when $h_e^2$ is relatively large, the FDR performance of MAAT is comparable to that of other methods (Fig. 3E–H).

### MAAT improves the expression imputation performance and identifies more gene-trait associations

We applied MAAT, along with four other widely used TWAS methods, PrediXcan, T-GEN, TIGAR, and EpiXcan, to detect significant genes associated with eight psychiatric traits, including Alzheimer's disease (AD), anorexia nervosa, bipolar disorder, depression, intelligence, insomnia, Parkinson's disease (PD), and schizophrenia. Initially, an imputation model was trained for each gene using the Religious Orders Study and Rush Memory Aging Project (ROS/MAP) reference panel [27, 28]. After data preprocessing (Methods section), we retained 10,940 gene imputation models for subsequent analysis, where the the gene expression levels were adjusted for various confounding effects using linear regression before the imputation step (Methods section).

To assess the performance of imputation $R^2$ for the three methods, after gene expression imputation on the ROS/MAP data, we conducted independent validations from the GTEx V8 database across 13 brain tissues (Methods section). Given that genes with low expression heritability often exhibit significantly larger causal effect sizes on complex traits [29], we set the threshold for test $R^2$ at 0.005. As illustrated in Additional file 1: Figs. S4 and S5, each method demonstrated a relatively balanced performance across different brain tissues. On average, MAAT obtained an average of 7354 genes with test $R^2 > 0.005$, compared to 4367 genes by PrediXcan, 4518 genes by T-GEN, 5622 genes by EpiXcan, and 6078 genes by TIGAR, suggesting a consistent increased power in prediction $R^2$ of MAAT. If a gene has a validation $R^2 > 0.005$ in more than half of the brain tissues (6 out of 13) in the GTEx V8 database, we consider it an imputable gene and proceed to the second step of association analysis.

After the imputation step of TWAS, the effect size estimation of cis-SNPs was obtained. We then applied FUSION [7] to leverage summary-level GWAS data for identifying gene-trait associations (Methods section). The detailed information of the eight GWAS summary database associated with eight traits is given in Table S1. Overall, as shown in Fig. 4A, MAAT identified 355 significant associations across the eight psychiatric traits (we set the $p$ value threshold to $5 \times 10^{-6}$, which is an approximation of the Bonferroni-corrected significance threshold $0.05/10940 = 4.57 \times 10^{-6}$ based on 10,940 genes, Additional file 1: Figs. S6–S21). In comparison, PrediXcan, T-GEN, EpiXcan, and TIGAR identified 81, 120, 168, and 278 significant associations, respectively. Among the eight traits, MAAT and TIGAR demonstrated superior performance in capturing genes related to anorexia nervosa, bipolar disorder, intelligence, PD, and schizophrenia compared to PrediXcan, T-GEN, and EpiXcan (Fig. 4, Additional file 1: Figs. S22–S28). For intelligence and schizophrenia, MAAT demonstrated significantly superior performance compared to TIGAR. Besides, all five methods yielded a higher number of significant findings compared to the other traits.

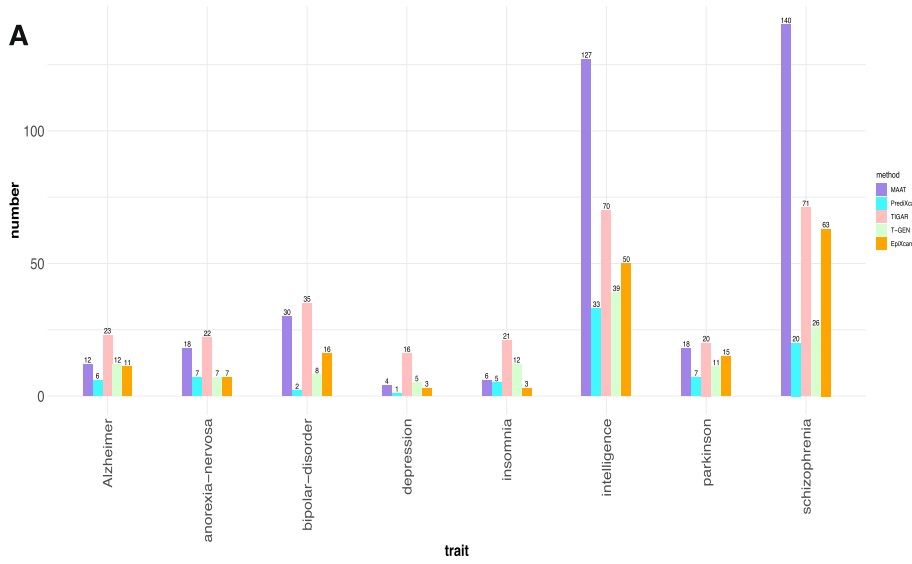

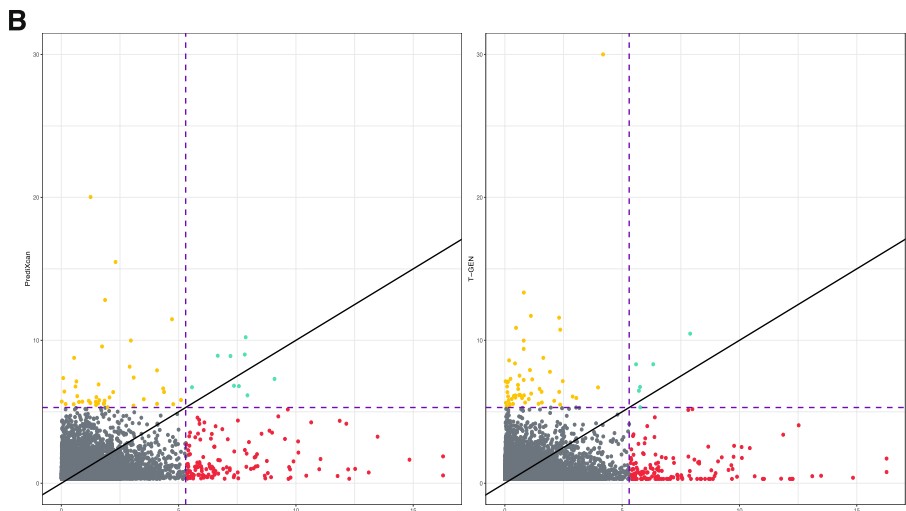

**Fig. 4** **A** The number of significant genes (***p*** value**< 5 × 10⁻⁶**) selected by MAAT, PrediXcan, T-GEN, TIGAR, and EpiXcan in eight traits. After fitting the imputation model in ROS/MAP dataset using MAAT, PrediXcan, T-GEN, TIGAR, and EpiXcan, we perform the association test in eight psychiatric traits, with the GWAS summary statistics obtained from Psychiatric Genomics Consortium. The *x*-axis refers to eight psychiatric traits and the *y*-axis refers to the number of significant genes ($p$ value $< 5 \times 10^{-6}$) selected by three methods. **B** Scatterplot of ***p*** values for gene-trait associations comparing MAAT with PrediXcan (left) and T-GEN (right) in schizophrenia GWAS data. Each dot represents a gene. The *x*-axis refers to the − log 10 ($p$ value) indicating the TWAS significance level calculated by MAAT, and the *y*-axis refers to the − log 10 ($p$ value) indicating the TWAS significance level calculated by PrediXcan (left) or T-GEN (right). The purple dashed line corresponds to the $p$ value cutoff of $5 \times 10^{-6}$. Genes not reaching the significance level in both methods are represented with gray dots, genes reaching the significance level in MAAT while not reaching the significance level in other methods are represented with red dots, genes reaching the significance level in both methods are represented with cyan dots

We further utilized the corresponding summary-level statistics for the analyzed eight psychiatric traits to estimate the distribution of SNP effect sizes. Based on GEN-ESIS [30], the effect size distribution is characterized by the proportion of underlying susceptibility SNPs and a mixture normal model for their effects. As shown in

Table S2, we found that traits with larger effect sizes of causal SNPs tend to possess a greater number of TWAS significant genes. For example, MAAT found more TWAS significant genes in intelligence, anorexia nervosa, and schizophrenia, which exhibit larger heritability as well. For PD, which is estimated to have a lower proportion of causal SNPs and heritability, MAAT still outperformed three methods in identifying more TWAS significant genes. To validate the reliability of the significant gene-trait associations identified by five methods, we utilized a combination of the Online Mendelian Inheritance in Man (OMIM) database [31] and the NHGRI-EBI GWAS catalog database [3] as a silver standard. As shown in Additional file 1: Fig. S29, except for AD and anorexia nervosa, MAAT consistently ranks among the top two methods in terms of AUC across the other six traits (with TIGAR also possessing relatively high AUC values). However, it is notable that all methods exhibit relatively low AUC values across different traits. Furthermore, as noted by previous research [32], TWAS results need further refinement due to the LD-hitchhiking effect. To address this, we conducted a colocalization analysis using fastENLOC [33] (Methods section). As shown in Additional file 1: Fig. S30, the ENLOC results reveal that many TWAS significant genes do not exhibit high gene-level colocalization probability (GLCP). This aligns with the findings from previous study [34], which highlighted that TWAS and colocalization analysis, as two integrative tools, often yield notably different results. However, MAAT, EpiXcan, and TIGAR have shown more conceptual replication [34] compared with PrediXcan and T-GEN. We further conducted pathway enrichment analysis on the MAAT significant genes with high GLCP values. We found that the enrichment levels of some key pathways were enhanced, suggesting that colocalization analysis can effectively filter out certain false positives in TWAS. However, the enrichment levels of other important pathways were reduced, due to the low GLCP levels of some significant TWAS genes [35–38]. A more detailed explanation of this analysis is provided in Additional file 1.

### Shared significant genes in multiple traits

Previous studies have demonstrated that genetic etiologies are shared among many neuropsychiatric disorders [39, 40]. Moreover, genetic correlations across many psychiatric phenotypes can be considerably high [41, 42]. In the findings returned by MAAT, we also found some significant genes exhibiting pleiotropic roles. For example, as shown in Fig. 5, MAAT identified nine genes (*SPI1*, *WDR6*, *ARL13B*, *HDGFRP3*, *PES1*, *PFKFB4*, *LRRC37A*, *NSF*, and *TCTA*) with high association levels ($p \leq 5 \times 10^{-6}$) in three out of eight traits. Previous studies have shown that *ARL13B* regulates the migration and placement of interneurons in the developing cerebral cortex [43]. Additionally, the expression of *ARL13B* variants underlies the neurological defects in Joubert syndrome patients [44]. Other genes such as *SPI1*, *LRRC37A*, *NSF*, and *TCTA* have also been reported to have effect on neurodegenerative diseases [45–48].

For some well-known highly correlated trait pairs, such as bipolar disorder and schizophrenia, schizophrenia and intelligence, and anorexia nervosa and depression, MAAT also identified some important genes playing roles in both traits. Specifically, 12 genes demonstrated high association levels in both bipolar disorder and schizophrenia, accounting for more than 25% of the total significant genes in bipolar disorder. Among

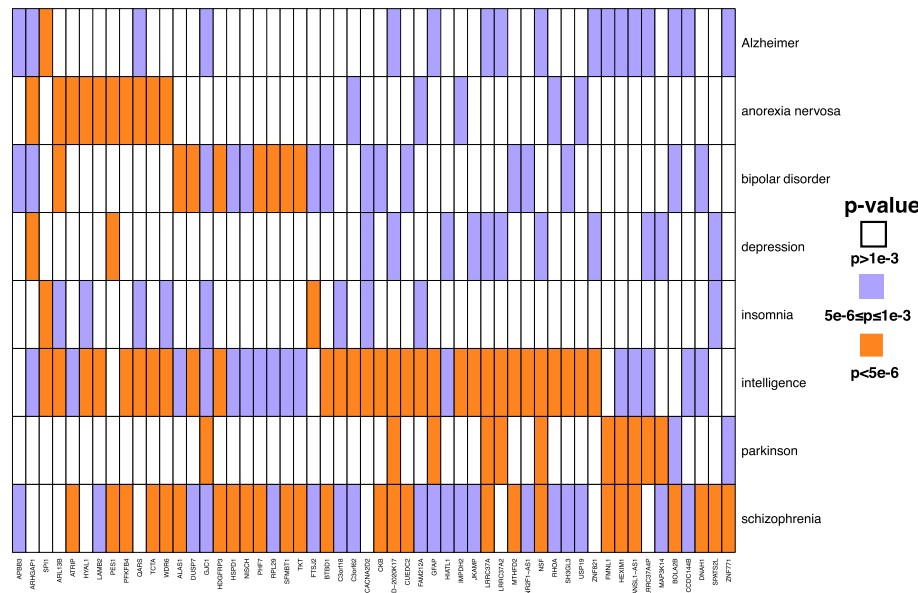

**Fig. 5** TWAS-significant genes selected by MAAT in eight psychiatric traits. Each column is a gene which has small *p* values in multiple psychiatric traits. Each row represents a psychiatric trait. The color of each box indicates the TWAS significance level of a gene in a psychiatric trait. The significance levels are divided into three subgroups (*p* value > $10^{-3}$, $10^{-6} \leq p$ value$\leq 10^{-3}$, and *p* value $< 10^{-6}$), which are denoted with different colors

these 12 genes, *SEMA3G*, *ALAS1*, *SFMBT1*, and *PHF7* flow into the NHGRI-EBI GWAS catalog [49] for both bipolar disorder and schizophrenia. In addition, *CNNM4* and *SF3B1* also belong to the bipolar disorder GWAS catalog, reinforcing the confidence level of MAAT's results. Regarding intelligence and schizophrenia, MAAT identified 21 genes with high association levels in these two traits, where the significant function of *ACTR1A*, *C22orf46*, *CKB*, *CYP2D6*, *LRRC37A*, *NCK1*, and *ZMAT2* have been well validated in large-scale GWAS studies. Notably, aside from intelligence and schizophrenia, *LRRC37A* also plays important roles in other psychiatric diseases. Previous studies have emphasized that *LRRC37A/2* contributes to the association between the 17q21.31 locus and PD through its interaction with *α*-synuclein and its effects on astrocytic function and inflammatory response [46], where the 17q21.31 locus has been shown to be genetically associated with an increased risk of *APOE* *ε*4-negative AD [50]. For the anorexia nervosa and depression pair, despite the limited number of significant genes in both disorders, MAAT identified two genes, *ARHGAP1* and *PES1*, that exert a significant impact on these two diseases. The pivotal role of *ARHGAP1* in anorexia nervosa has been firmly established [51].

To better interpret the identified gene-trait associations and to understand the potential pathogenic mechanisms shared among multiple traits, we utilized the STRING protein-protein association network database [52] to explore the significant genes identified by MAAT. As shown in Additional file 1: Fig. S31, almost for every pair of traits, there exists strong interactions among their TWAS significant genes, which aligns with existing studies suggesting shared genetic risk factors across multiple psychiatric conditions. Several genes, such as *PES1*, *SPI1*, and *NSF*, exhibit multiple significant associations across different traits, indicating their potential central roles in the genetic

network influencing psychiatric disorders. For instance, *PES1* shows associations with anorexia nervosa, depression, and schizophrenia, and has strong interactions with genes like *WDR74*, *RBM28*, and *RRP7A*. This suggests that *PES1* could be a key player in the pathogenesis of these conditions, acting as a hub that links various genetic pathways.

**Shared enriched pathways in multiple traits**

To further increase our understanding of the underlying mechanisms of the significant genes selected by MAAT, we first performed pathway enrichment analysis with respect to each trait. Top 50 significant pathways for each trait are shown in Additional file 1: Figs. S32–S39. Numerous findings are well confirmed by existing studies. For example, the enrichment of fructose and mannose metabolism pathways in anorexia nervosa [53]; rough endoplasmic reticulum pathway in bipolar disorder [54]; the enrichment of cellular response to corticotropin-releasing hormone stimulus pathway, cerebellar cortex development pathway, and NSL complex pathway in PD [55]; the enrichment of dendritic spine membrane pathway and endoplasmic reticulum to Golgi vesicle-mediated transport pathway in schizophrenia [56, 57], etc. Besides, some KEGG disease pathways were also selected in corresponding diseases, such as bipolar disorder pathway in bipolar disorder ($p = 9.35 \times 10^{-16}$), Parkinson's disease pathway in PD ($p = 3.53 \times 10^{-8}$), and schizophrenia pathway in schizophrenia ($p = 2.07 \times 10^{-21}$), suggesting the power of MAAT in capturing the most relevant pathways.

Furthermore, upon comprehensive analysis of the top 50 significant pathways across all eight traits, as shown in Fig. 6A, we identified 66 pathways that exhibited significant enrichment in more than three traits (adjusted $p$ value $\leq 0.05$). Notably, MAAT identified numerous significant pathways associated with neurological disorders. These encompass nervous system diseases, neural tube patterning, NFAT protein binding, positive regulation of synaptic transmission (GABAergic), response to lithium ion, Rho protein signal transduction, schizophrenia, small GTPase mediated signal transduction, and smoothened signaling pathway. Of particular significance, six pathways demonstrated enrichment in four traits. Noteworthy among them is the early endosome pathway, which exhibited enrichment in bipolar disorder, depression, intelligence, and schizophrenia. This pathway's pivotal role in conditions like AD and PD has already gained widespread recognition [58, 59]. Consequently, the results from MAAT underscore the early endosome pathway as a quintessential exemplar of a pleiotropic pathway.

**Gene-trait associations assigned with transcription factor-related annotation**

We identified 75 genes whose dominating disease-risking mechanism flowed into the annotation of TF (for convenience, we call these 75 genes as TF-tagged genes) in the analyzed eight traits. To follow-up on these findings, we combined the TF-disease database and TF-gene regulatory network to explore the possible mode of action for those TF-tagged genes (Methods section). As shown in Table S3, among the four traits possessing TF-tagged genes, many of the TF-tagged genes are found to be regulated by TFs which are well-validated to play important roles in the corresponding trait. For instance, TF-tagged genes *CRABP1*, *DOK1*, *RBM28*, and *C12orf65* are regulated by multiple schizophrenia-related TFs. Among them, previous studies have shown that Crabp1 knockout (CKO) mice exhibited reduced anxiety-like behaviors

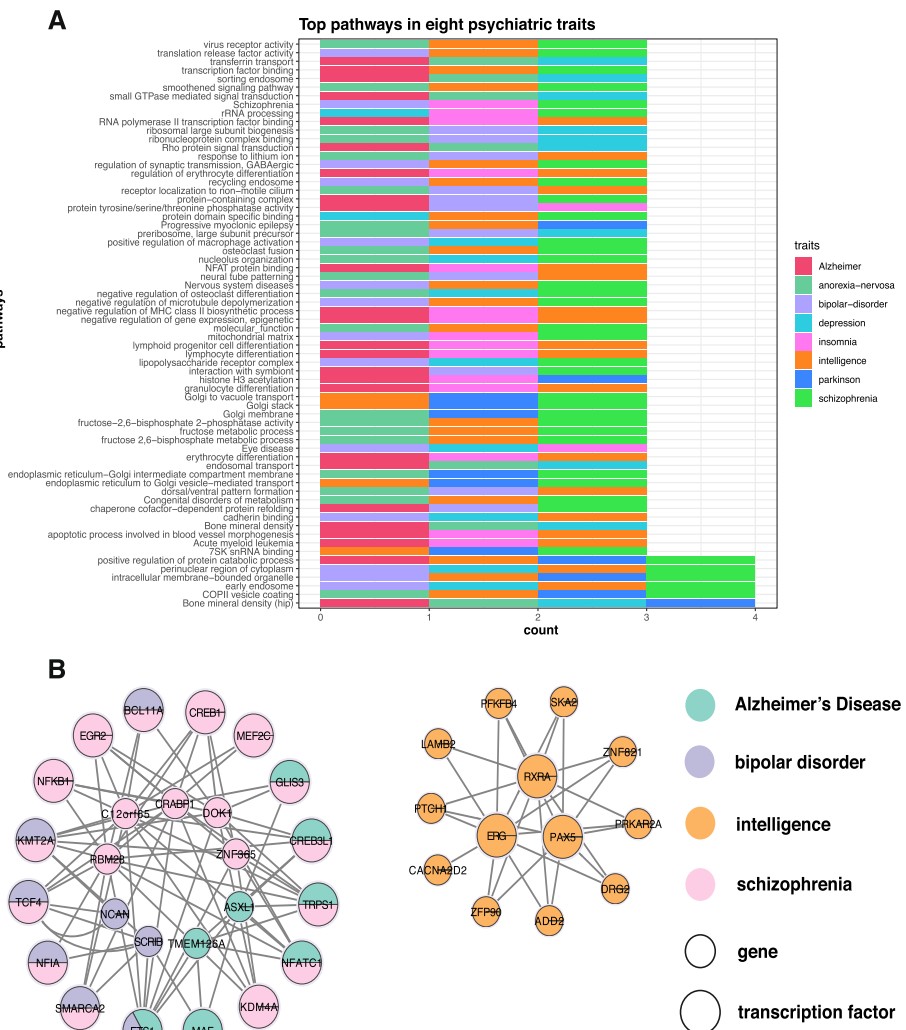

**Fig. 6  A** Top pathways enriched in eight psychiatric traits. For each of the eight psychiatric trait, pathway enrichment analysis is performed in the gene set with high significance level ($p$ value $< 5 \times 10^{-6}$) calculated by MAAT. Pathways which are enriched (adjusted $p$ value $< 0.05$) in more than two traits are listed. Each row represents a pathway which is enriched in multiple traits. Each block is a pathway-trait enrichment association. The eight psychiatric traits are denoted with different colors. **B** TF-gene regulation network. Each circle represents a gene or transcription factor, where small circles represent TWAS significant genes selected by MAAT ($p$ value $< 5 \times 10^{-6}$) whose most important annotation is transcription factor, and big circles represent transcription factors which are well-validated to play important roles in the corresponding traits. Each line is a regulatory relationship between TF and gene. TWAS significant genes or well-validated TFs playing roles in different psychiatric traits are filled with different colors. The pie chart is adopted to display genes or TFs which play functions in multiple traits

accompanied by a lowered stress induced-corticosterone level [60]. *CRABP1* has also been recognized as a biomarker in defining schizophrenia organoids [61]. By our calculation, the regulation of *CRABP1* by certain schizophrenia-associated TFs may constitute a crucial avenue through which it exerts its influence on schizophrenia. For example, TFs *SMARCA1*, *TCF4*, and *TRPS1* have been confirmed to play crucial roles in schizophrenia, and they have been confirmed to target *CRABP1*.

To further reveal possible pleiotropic mechanisms for TF-gene regulation in TWAS, we merged all trait-specific TF-gene regulatory networks together. As illustrated in Fig. 6B, the comprehensive TF-gene regulatory network primarily comprises two sub-networks. One encompasses intelligence-related TFs and their target genes, while the other is relevant to the regulatory network associated with AD, bipolar disorder, and schizophrenia. Certain TFs such as *ETS1*, *MAF*, *SMARCA2*, *TRPS1*, and *CREB3L1* appear to modulate TF-tagged genes in multiple traits, which could be key elements linking AD, bipolar disorder, and schizophrenia through shared regulatory pathways.

### Gene-trait associations assigned with epigenetics-related annotation

We found 121 genes whose dominating disease-risking mechanism originates from epigenetic changes. For convenience, we call these genes as epigenetic-tagged genes. As shown in Tables 1 and S4, among the eight traits, schizophrenia has the most epigenetic-tagged genes. In order to validate the reproducibility of the epigenetic annotation assignment, we searched the EWAS catalog dataset [62]. It can be shown that among the eight traits, schizophrenia has the most 18 epigenetics-tagged genes which have been validated by EWAS studies, such as *CACNA1C* and *TCTA* (Table 1).

MAAT also found that some genes, such as *SPI1*, *HEXIM1*, *KCNC3*, and *ARHGAP1*, which not only showed high significance level in multiple traits, but the potential mechanism through which these genes affect multiple traits can be attributed to epigenetic factors. Specifically, MAAT found that *SPI1*, which is specifically expressed in microglia in the brain, may have an impact on both AD and intelligence via epigenetic access, and this has been validated by existing EWAS studies in both AD and intelligence. Recent evidence from GWAS suggests that reductions in *SPI1* contribute to a delayed onset of AD [63]. Furthermore, the SPI1-dependent transcriptional pathway could drive the

**Table 1** Epigenetic-tagged genes in eight psychiatric traits

| Traits | Genes with epigenetics annotation |
| --- | --- |
| Alzheimer's disease | **SPI1**, *LYPD5, TSC22D4* |
| Anorexia nervosa | *ARHGAP1, HYAL1* |
| Bipolar disorder | *KCNC3* |
| Depression | **ARHGAP1**, *TMEM141* |
| Insomnia | *LTN1* |
| Intelligence | **ADCY5**, **ANKRD9**, **C12orf65**, **CPD**, **IP6K1**, **SESN1**, **SPI1**, **TMCO6**, **UBA7**, *BTBD1, FAM212A, GOSR2, HDGFRP3, RRP7A* |
| Parkinson's disease | *GOSR2, HEXIM1* |
| Schizophrenia | **ABHD14A**, **ALDH3A2**, **AP1G1**, **ATPAF2**, **BDH2**, **CACNA1C**, **CNNM4**, **FAM219A**, **FLII**, **FMNL1**, **HAPLN4**, **MTHFD2**, **NHP2L1**, **PTGES2**, **RPRD1B**, **SPATS2L**, **TCTA**, **ZSWIM6**, *ACTR1A, AEN, BOLA2B, BTBD1, CA8, CALHM2, CASC4, COQ5, CUEDC2, EFTUD1, GBF1, GCC1, GLTP, HAR1A, KCNC3, KIF3C, LRR37A, PSD, RRP7A, TCTN2...* |

Each row corresponds to a psychiatric trait along with its epigenetic-tagged genes reaching the significant level ($p$ value $< 5e - 6$). Genes in bold are validated by existing EWAS studies. Only part of significant epigenetic-tagged genes in schizophrenia are shown for saving space

interleukin (IL)-33-induced epigenetic and transcriptional regulation of microglial state transition, thereby enhancing beta-amyloid (Aβ) clearance and alleviating AD pathology [64]. In addition, HEXIM1 is an epigenetic-tagged gene in both PD and schizophrenia. Previous studies validated that HEXIM1, the tumor suppressor, can be regulated by the H3K4me3/2 demethylase *KDM5B*. Therefore, *KDM5B* is identified as a druggable target to inhibit the proliferation of cancer by upregulation the expression levels of *HEXIM1* [65]. *ARHGAP1* is an epigenetic-tagged gene in both anorexia nervosa and depression, and this mechanism has been validated by EWAS studies in depression. Based on the fact that genetic factors substantially contribute to the observed comorbidity between anorexia nervosa and major depression [66], we have compelling reasons to believe that epigenetic factors play a pivotal role in the process where the *ARHGAP1* gene influences anorexia.

### Gene-trait associations assigned with conservation-related annotation

As another annotation used in our analysis, conservation means evolutionary stability for particular genes. Conserved genes usually play a key role in essential biological process of the cell, and mutation or variation on a conserved gene is more likely to be pathogenic. As detailed in Table 2, MAAT identified 97 gene-trait associations assigned with the conservation label. For simplicity, we term these genes as conservation-tagged genes. We validated the rationale behind conservation-tagged genes by cross-referencing them with the housekeeping gene set. This validation is grounded in the rationality that housekeeping genes exhibit conserved functions and expression patterns [67].

In the realm of AD, there are a total of ten conservation-tagged genes. Among them, four have been validated as housekeeping genes. Within the remaining six genes, *PTPMT1* encodes a protein tyrosine phosphatase localized to the mitochondrion and prevents intrinsic apoptosis probably by regulating mitochondrial membrane integrity. Although *PTPMT1* has not been classified as a housekeeping gene, previous studies have demonstrated that its function is evolutionarily conserved across a spectrum spanning

**Table 2** Conservation-tagged genes in eight psychiatric traits

| Traits | Genes with conservation annotation |
|---|---|
| Alzheimer's disease | **ARHGAP1**, **DDB1**, **DNAJC7**, **GIGYF1**, *CREBZF, GIGYF1, PTPMT1, RPL32P3, SMG9, ZNF283* |
| Anorexia nervosa | **TCTA**, *LAMB2, SLC25A20* |
| Bipolar disorder | **KAT8**, **MSL1**, **RPL29**, *CLEC18A, GJC1, HARS, NDFIP2, NFXL1, PELI1, PHF7, RPL29, TKT, WDR74* |
| Depression | *PMS2P3, TLR4* |
| Insomnia | *FTSJ2* |
| Intelligence | **ADD1**, **APEH**, **IP6K2**, **NDUFA2**, **QRICH1**, **RHOA**, **RPL18A**, **SHISA5**, **SMDT1**, **THAP11**, *ARL13B, ARPP21, ATRIP, BSN, C16orf86, C1QTNF4, C3orf18, C9orf129, CCDC51, FAM193A, GTDC1, IFRD2, IMPDH2, KBTBD4, KIAA1841, MTHFD2, NCKIPSD, NICN1, PDPR, RBBP4 RILPL1, SULT1A1, SULT1A4, USP19* |
| Parkinson's disease | *CRHR1, LRRC37A4P, MAP3K14* |
| Schizophrenia | **CREB3**, **MAFK**, **PSMG3**, **SF3B1**, **TMED2**, **TWF2**, *ATRIP, BRAP, C2orf69, C7orf50, GPER1, HAUS2, IDH3A, KREMEN1, MICALL2, NCK1, NIT2, OSCP1, PHF7, PITPNM2, RFT1, RILPL1, SH3D21, SNX8, SPAG8, TKT, TMEM62, TYW5, UBR1, USP32P2, USP32P3, ZMYM4* |

Each row corresponds to a psychiatric trait along with its conservation-tagged genes reaching the significant level
(*p* value $< 5e-6$). Genes in bold are known housekeeping genes

from bacteria to mammalian cells [68]. Intriguingly, recent work has linked it to both AD and glaucoma in a GWAS study [69].

In the context of schizophrenia, MAAT identified a total of 32 conservation-tagged genes. Among them, *GPER1* encodes a membrane estrogen receptor, with its physiological function established as conserved in vertebrates [70]. In a study leveraging an MK-801-induced mouse model of schizophrenia, *GPER1* exhibited significant impact on cognitive, learning, and memory functions, thereby suggesting a potential role in the pathogenesis of schizophrenia [71]. Notably, for another conservation-tagged gene *PSMG3*, we calculated that among its seven annotations, there were seven pairs of annotations displayed negative correlations. Partially because of its intricate annotation profiles, the other TWAS methods failed to detect the significance of *PSMG3*. Interestingly, in a recent study investigating schizophrenia from an evolutionary perspective, the SNP rs3800926, located in a human accelerated region (HAR), was validated to play function in schizophrenia, and this SNP is mapped to *PSMG3* [72].

In addition, *SMG9* in AD [73], TCTA in anorexia nervosa [74, 75], *RHOA* and *IMPDH2* in intelligence [76, 77], *CRHR1* in PD [78], and *PHF7* in schizophrenia [79] are all reported to be conserved among a wide range of species, although their relations with the corresponding traits remain to be explored.

### Gene-trait associations assigned with annotation of proximity to TSS/TES/coding

Numerous studies have suggested that SNP loci in close proximity to TSS/TES/coding regions may exert influence on key regulatory elements of genes, such as promoters and enhancers, thereby modulating gene expression levels. Among those genes whose most important annotation flowing into "promixity to TSS/TES/coding," through comparison with GWAS catalog, we identified several significant SNPs close to the TSS/TES/coding regions of these genes. This to some extent confirms that the mechanism by which genes affect traits may be attributed to the proximity of vital SNPs to TSS/TES/coding regions.

For instance, as shown in Fig. 7, in the context of PD, our computations revealed that the mechanisms of genes *KANSL1* and *NSF* are intricately linked to their proximity to coding regions. The SNP rs58879558, located in the MAPT region, exhibited associations with both neuroticism and PD [80], being mere 12K bp away from the starting point of a exon of *KANSL1*. Similarly, the SNP rs183211 is only 20 bp away from the starting point of an exon of gene *NSF*. Recent studies have reported that this SNP is associated with both ovarian cancer and PD. Furthermore, in the positive genetic correlation between PD and ovarian cancer, the association is primarily driven by rs183211 [81].

In the realm of intelligence, we calculated that the mechanism through which *ARFGEF2* influences intelligence is correlated with its proximity to TSS/TES. The SNP rs6095360 on this gene has been reported to correlate with intelligence [82], and its distance from the transcription start site of *ARFGEF2* is only about 6 kb.

In the case of schizophrenia, we observed a concentration of numerous TWAS significant genes in the region spanning 52–54 Mb on chromosome 3. The mechanisms underlying the influence of *ALAS1*, *DNAH1*, and *PRKCD* on schizophrenia are attributed to their proximity to TSS/TES. As for the *PRKCD* gene, it has been reported to exert an influence on schizophrenia [83]. Existing GWAS analyses have further reported that

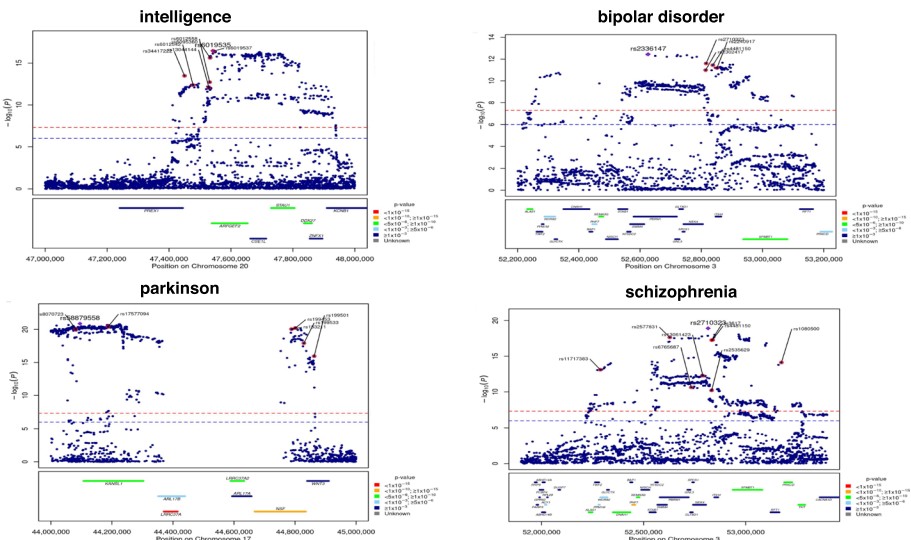

**Fig. 7** LocusZoom plots for some genes whose most important annotations are proximity to TSS/TES/coding region in four psychiatric traits. For each sub-figure, the upper plot represents the GWAS significance level for SNPs in a genomic region for a specific psychiatric trait, with each blue dot being a SNP. The x-axis denotes the location in a chromosome, the y-axis is the − log 10 (p value) representing the significance level for a SNP, where the significance level is obtained from the GWAS-summary database. The marked SNPs with red dots are discovered by existing GWAS studies. The lower plot in each sub-figure represents the TWAS significance level for several genes calculated by MAAT, genes with different significance levels are plotted with different colors

the SNP rs1080500 plays a role in both schizophrenia and attention deficit hyperactivity disorder (ADHD), with a proximity of only about 20 kb to *PRKCD*. Additionally, SNP rs1080500 also serves as an eQTL locus for *GNL3*, a gene known for its role in regulating neuron differentiation [84].

## Discussion

We propose MAAT framework to harness the comprehensive effects of different annotations on genomic function in TWAS. MAAT integrates diverse SNP annotations far beyond epigenetic aspect and employ a PPMx model to allocate similar effect sizes for cis-SNPs with similar annotation profiles. Through extensive simulation studies, we demonstrate that MAAT exhibits greater imputation $R^2$ and increased TWAS power to implicate associations, while maintaining low type I error rates across various settings. Applying MAAT to eight psychiatric diseases, more gene-trait associations have been identified compared to existing methods. Furthermore, MAAT goes beyond traditional TWAS approaches and assigns the most important annotation for each significant gene-trait association, providing a more nuanced understanding of the genetic basis of complex diseases.

Following downstream molecular biology analysis, MAAT demonstrates that genetic etiologies are commonly shared across various neuropsychiatric disorders. The interconnectedness between these diseases is evident not only in the substantial overlap of significant TWAS genes but also, upon an angle-based annotation assignment, in the observation that the mechanisms through which genes influence

different psychiatric diseases converge on shared ways. For instance, the influence of *KCNC3* on both bipolar disorder and schizophrenia may manifest through epigenetic mechanisms.

However, MAAT also has its limitations. First, the computational burden of the PPMx is relatively heavy because the parameter estimation is achieved by Markov Chain Monte Carlo (MCMC). For a gene with approximately 3000 cis-SNPs, training the imputation model requires about 2 CPU hours on one core. Bayesian acceleration algorithms are deserved to be explored in the following studies. Second, although many genes along with their influencing mechanisms have been verified by means of some existing database, statistical tests or standardized criteria for the annotation assignment process are deserved to be developed to enhance reliability and interpretability.

Looking ahead, with the integration of new data types and biological contexts, there are still numerous unexplored avenues for MAAT. First, with the continued evolution of single-cell technologies, there is a compelling opportunity to extend TWAS into the realm of single-cell transcriptomics. There have already been attempts to integrate GWAS with single-cell studies [85]. Traditional TWAS models are built on tissue-level bulk data, potentially obscuring cell-specific effects. By dissecting gene expression patterns at the single-cell level, researchers can gain unprecedented insights into cell type-specific regulatory mechanisms. This expansion into single-cell TWAS holds the potential to uncover novel therapeutic targets and enhance our understanding of cell type heterogeneity in complex diseases.

Second, ethnic diversity remains a critical consideration in TWAS. Broadening studies to encompass diverse populations can unveil population-specific regulatory architectures and disease mechanisms. This inclusive approach can lead to more equitable and effective precision medicine strategies that account for the genetic diversity within global populations.

Third, recent studies have demonstrated the effectiveness of cross-tissue TWAS [11, 86]. In our single-tissue model incorporating annotation information, we utilized the informative annotation profiles from FAVOR. If we were to incorporate annotation information in a cross-tissue TWAS framework, the inclusion of tissue-specific annotation data is bound to provide higher accuracy in exploring the etiology of diseases. Building a new statistical model to combine multiple tissues while accounting for tissue-specific annotation is also a valuable direction for future exploration.

Finally, in terms of enhancing the reliability of TWAS results through further colocalization analysis, due to the relatively conservative nature of existing colocalization methods, coupled with the fact that TWAS and colocalization methods often yield very different results [34], it is essential to develop TWAS fine-mapping methods to enhance the credibility of TWAS findings. While FOCUS is another powerful tool for fine-mapping TWAS gene sets [32], previous study [87] demonstrated that, in the presence of within-gene LD hitchhiking, the InSIDE assumption (instrument strength independent of direct effect) is violated, leading the FOCUS model to inflate false positives when finding causal genes [34, 88]. Therefore, combining existing TWAS fine-mapping methods with locus-level colocalization is a promising direction for future research.

## Conclusions

In this paper, we introduce a novel approach MAAT to integrate various annotation information into the TWAS framework, which improves the imputation $R^2$ and increases TWAS power to implicate associations. Unlike depending on the assumption of a positive correlation between the cis-SNPs' effect size and the annotation score, MAAT utilizes a PPMx prior to group cis-SNPs based on the distribution of annotations. This induced grouping enables cis-SNPs with similar annotation distributions to share comparable effect sizes. Furthermore, MAAT employs an angle-based metric to pinpoint the most impactful annotation for each significant gene-trait association. Application of MAAT to eight psychiatric traits uncovered more significant gene-trait associations compared to existing state-of-the-art TWAS methods. Our results reveal that many genes not only exhibit pleiotropic effects across multiple psychiatric traits but also, based on annotation assignments, suggest that the mechanisms by which these genes affect different traits may be similar.

## Methods

### General two-step TWAS

General TWAS methods typically consist of two steps: the imputation step and the association step. In the imputation step, individual-level data from a reference panel is employed to impute the expression profile of each gene using the genotype information of cis-SNPs. For each gene $g$, we consider the following model:

$$\boldsymbol{E}_g = \boldsymbol{X}\boldsymbol{\beta} + \boldsymbol{\epsilon}, \ \boldsymbol{\epsilon} \sim N(0, \sigma_e^2 \mathbf{I}_n).$$

Where $\boldsymbol{E}_g$ is an $n$-vector denoting the expression profiles of gene $g$ across $n$ individuals, $\boldsymbol{X}$ is a gene-specific $n \times p$ genotype matrix, where each column contains the genotype information for a cis-SNP of gene $g$. In our study, we incorporate cis-SNPs within a 1 Mb range of the gene's transcriptional start and end sites to construct $\boldsymbol{X}$. $\boldsymbol{\beta}$ is the effect size vector, and $\boldsymbol{\epsilon}$ is the residual error, with each element following a Gaussian distribution with mean 0 and variance $\sigma_e^2$. We drop the intercept term for assuming $\boldsymbol{E}_g$ and $\boldsymbol{X}$ are standardized.

In the second association step, by leveraging the estimated effect size $\hat{\boldsymbol{\beta}}$ in the first step, the genetically regulated expression (GReX) for independent GWAS samples can be imputed. Subsequently, a gene-based test is performed by testing the association between the imputed $\widehat{\mathbf{GReX}}$ and the phenotype of interest. When individual-level GWAS data are available, suppose $\boldsymbol{X}_{\text{new}}$ is the genotype matrix for gene $g$ in the new GWAS samples, the GReX is imputed by $\widehat{\mathbf{GReX}}_g = \boldsymbol{X}_{\text{new}}\hat{\boldsymbol{\beta}}$. Then, a generalized linear model (GLM) between $\widehat{\mathbf{GReX}}_g$ and phenotype $\boldsymbol{Y}$ is adopted to perform the association test:

$$E[\boldsymbol{Y}|\widehat{\mathbf{GReX}}_g] = g^{-1}(\widehat{\mathbf{GReX}}_g \omega + \boldsymbol{C}_{\text{new}}\alpha),$$

where $g$ is the link function and $\boldsymbol{C}_{\text{new}}$ is the covariant matrix. For a quantitative phenotype $\boldsymbol{Y}$, $g$ can be the identity function; for a dichotomous phenotype $\boldsymbol{Y}$, $g$ can be set as the logit function. The association between $\widehat{\mathbf{GReX}}_g$ and the phenotype is evaluated by testing $H_0 : \omega = 0$ versus $H_1 : \omega \neq 0$. When only summary-level GWAS data are available, we employ the burden $Z$-score implemented in FUSION [7] to identify significant $\widehat{\mathbf{GReX}}_g$-phenotype associations. Specifically, the burden $Z$-score is defined as:

$$Z = \frac{Z_{SG}^T \hat{\boldsymbol{\beta}}}{\sqrt{\hat{\boldsymbol{\beta}}^T V \hat{\boldsymbol{\beta}}}},$$

where $Z_{SG}$ is the vector of $Z$-scores for all cis-SNPs from summary-level GWAS data, $\hat{\boldsymbol{\beta}}$ is estimated from the imputation step, $V$ is the covariance matrix for all cis-SNPs of gene $g$, which can be estimated from reference panel used in the imputation procedure or other reference databases such as 1000 Genomes Project [89] and UK Biobank data [90]. In MAAT, we estimate $V$ using UK Biobank data.

### Annotation incorporation by product partition model with covariates (PPMx)

We assume $\rho = \{S_1, S_2, \cdots, S_{k_p}\}$ is a partition of the $p$ cis-SNPs into $k_p$ clusters, $\boldsymbol{Z} = (z_1, \cdots, z_p)$ is a $p$-vector of latent variables, with $z_k = j$ if the $k$-th cis-SNP belongs to cluster $j$ under such partition, i.e., $k \in S_j$. Suppose there are $m$ annotation scores allocated to cis-SNP $k$, which are denoted as $\boldsymbol{W}_k = (W_{k1}, \cdots, W_{km})$. Therefore, the annotation matrix for all $p$ cis-SNPs can be represented by $\boldsymbol{W} = (\boldsymbol{W}_1^\top, \cdots, \boldsymbol{W}_p^\top) \in \mathbb{R}^{m \times p}$. Let $\boldsymbol{W}_j^* = \{\boldsymbol{W}_k : k \in S_j\} \in \mathbb{R}^{m \times |S_j|}$ denote the cluster $j$-specific annotation information, and let $\boldsymbol{W}_{jl}^*$, the $l$-th row of $\boldsymbol{W}_j^*$, denote a $|S_j|$-vector containing the $l$-th annotation information for cis-SNPs in cluster $j$, where $|S_j|$ is the cardinality of set $S_j$.

In order to simplify computation, following [8, 91, 92], we introduce a random effect term $\boldsymbol{u} \sim N(\boldsymbol{0}, \sigma_e^2 \sigma_0^2 \boldsymbol{K})$, where $\boldsymbol{K} = \boldsymbol{X}\boldsymbol{X}^\top / p$ is the genetic relatedness matrix (GRM). Therefore, $\boldsymbol{u}$ can be represented as $\boldsymbol{u} = \boldsymbol{X}\boldsymbol{\xi}$ with $\boldsymbol{\xi} \sim N(0, \sigma_e^2 \sigma_0^2 \mathbf{I}_p / p)$. We reformulate the original gene expression imputation model as:

$$\boldsymbol{E}_g = \boldsymbol{X}\boldsymbol{\beta} + \boldsymbol{\epsilon} = \boldsymbol{X}\tilde{\boldsymbol{\beta}} + \boldsymbol{u} + \boldsymbol{\epsilon}.$$

To leverage multiple annotation information in the imputation step, we utilize PPMx [22] to build a bridge linking the cis-SNPs' effect sizes and their annotation profiles:

$$
\begin{aligned}
&\tilde{\beta}_k | z_k = j, \sigma_e^2, \sigma_j^2 \sim N(0, \sigma_e^2 \sigma_j^2), \text{ for } k = 1, \cdots, p; \\
&\sigma_e^2 \sim \mathrm{IG}(a_e, b_e); \ \sigma_0^2 \sim \mathrm{IG}(a, b); \ \sigma_j^2 \sim \mathrm{IG}(a, b), \text{ for } j = 1, \cdots, k_p; \\
&P(\boldsymbol{Z} | \boldsymbol{W}) \propto \prod_{j=1}^{k_p} g(\boldsymbol{W}_j^*) C(S_j) \\
&g(\boldsymbol{W}_j^*) = \prod_{l=1}^{m} g(\boldsymbol{W}_{jl}^*); \ g(\boldsymbol{W}_{jl}^*) = \prod_{t \in S_j} \phi(W_{tl} | \hat{\mu}_{jl}, \hat{\sigma}_{jl}^2) \\
&C(S_j) = (|S_j| - 1)!
\end{aligned}
\tag{1}
$$

Here $\mathrm{IG}(a, b)$ denotes an inverse gamma distribution with parameters $a$ and $b$. $C(S_j) = (|S_j| - 1)!$ is a cohesion function to ensure that cis-SNPs with homogeneous annotation profiles are not excessively fragmented into small clusters (Additional file 1). $\phi(x | \mu, \sigma^2)$ is the density function of a normal distribution with mean $\mu$ and variance $\sigma^2$. $\hat{\mu}_{jl}$ and $\hat{\sigma}_{jl}^2$ are the sample mean and sample variance of $\boldsymbol{W}_{jl}^*$. $g(\boldsymbol{W}_j^*)$ is a similarity function. The higher homogeneity level of cis-SNPs in cluster $j$, the larger $g(\boldsymbol{W}_j^*)$ is [93]. In MAAT, the similarity function is a particular choice of the double

dipping similarity function [94]. A detailed derivation of $g(W_j^*)$ and some further explanations of the PPMx model are provided in Additional file 1.

It is worth noting that when a Dirichlet process (DP) prior is assumed for the distribution of cis-SNPs' effect size [8, 92], the marginal distribution that DP induces on partition $\rho$ (analogous to $Z$) is a product partition model (PPM) only with cohesion function $C(S_j) = M(|S_j| - 1)!$ [22, 95], where $M$ is the total mass parameter of the DP prior. Therefore, compared to the prior on $Z$ in MAAT where both the cohesion function and similarity function are included, the TIGAR method [8] can be regarded a special case of MAAT where annotation information is not considered.

By model (1), we assign a non-parametric prior on the allocation of $p$ cis-SNPs. The synergy of the similarity function $g$ and cohesion function $C$ ensures that cis-SNPs with similar annotation profiles are grouped into the same latent cluster and prevent from being splitted into small clusters. By assigning this flexible non-parametric prior, MAAT can adapt to a wide range of relationship architectures between effect size and annotation, jumping out of the widely-adopted linear assumption between annotation profile and effect size [10, 15, 16].

In practice, we develop a Markov Chain Monte Carlo (MCMC) sampling algorithm to obtain the posterior samples and achieve parameter estimation [96, 97]. Details of the MCMC sampling algorithm are shown in Additional file 1.

Given that the effect size for each cluster $j$, which can be indicated by $\sigma_e^2 \sigma_j^2$, reflects the importance level of different cis-SNPs, we propose a scoring mechanism to quantify the importance level for each cis-SNP based on the MCMC iteration results (Additional file 1). Consequently, MAAT can fine-tune $\beta$ to different sparsity levels by setting the effect size of less influential cis-SNPs to zero. In real data analysis, we investigate four sparsity levels for $\beta$—5%, 10%, 20%, and 30%. Each sparsity level of $\beta$ is put into the second association step, and the corresponding four $p$ values are combined using ACAT to yield the significance level of gene-trait association.

### Selection of functional annotations

We downloaded functional annotation data from FAVOR [14, 20], which facilicates multi-source variant functional information for all possible nine billion single nucleotide variants (SNVs) across the genome. FAVOR stores 160 functional annotation values spanning from integrative scores (such as aPC of conservation scores) to isolated-aspect annotation scores (such as chromatin states, mutation density), where the aPC refers to the first principal component of the set of individual annotation scores belonging to the same category [14], say, the epigenetic function category. In MAAT, considering the computational efficiency and the variance of annotation, we incorporated seven integrated functional annotations with relatively large variances to increase TWAS power, including aPC of conservation, aPC of epigenetics, aPC of mappability, aPC of transcription factor, aPC of proximity to coding region, aPC of proximity to transcription starting site (TSS) and transcription ending site (TES), and FATHMM-XF score for coding variants [21]. The SNP coordinates are converted from GRCh38/hg38 to GRCh37/hg19 reference assembly using LiftOver.

### Annotation assignment for genes

For a gene *g*, suppose the effect size that cis-SNPs impose on gene expression is denoted as $\boldsymbol{\beta}$, while we use $\boldsymbol{\alpha}$ to denote the effect size that cis-SNPs impose on phenotype. When the joint effect size $\boldsymbol{\alpha}$ is not accessible, we use the marginal effect size available from GWAS summary data to approximate $\boldsymbol{\alpha}$. Suppose $V$, the covariance matrix for all cis-SNPs of a specific gene, has the eigendecomposition $V = \boldsymbol{U}\boldsymbol{\Lambda}\boldsymbol{U}'$. We let $\boldsymbol{\beta}_r = \boldsymbol{\Lambda}^{\frac{1}{2}}\boldsymbol{U}'\boldsymbol{\beta}$ and $\boldsymbol{\alpha}_r = \boldsymbol{\Lambda}^{\frac{1}{2}}\boldsymbol{U}'\boldsymbol{\alpha}$ represent the rotated effect sizes.

Based on some basic calculation (see Additional file 1 for detailed information), we reach the conclusion that the mean of the TWAS *Z*-score is determined by the cosine distance between the rotated effect sizes $\boldsymbol{\beta}_r$ and $\boldsymbol{\alpha}_r$. If $\tilde{\boldsymbol{W}}_l = (W_{1l}, \cdots, W_{pl})^T$ denotes the *l*-th annotation information for *p* cis-SNPs, we assign the important annotation *l\** to each gene by the following fomula:

$$l^* = \underset{l=1,\cdots,L}{\operatorname{argmax}} \left| \cos(\tilde{\boldsymbol{W}}_{l,r}, \boldsymbol{\beta}_r) + \cos(\tilde{\boldsymbol{W}}_{l,r}, \boldsymbol{\alpha}_r) \right|,$$

where *L* is the number of annotations included in MAAT. In this study, we set $L = 7$.

### *p* value combination by ACAT

We apply ACAT to combine *p* values obtained from different sparsity levels of $\beta$ into an omnibus *p* value. In MAAT, we set four sparsity levels, i.e., the proportions of causal SNPs are set to be 5%, 10%, 20%, and 30%. Concretely, let $p_i$ be the *p* value from the *i*-th sparsity level of $\beta$, the Cauchy combination test statistic is defined as:

$$T = \sum_{i=1}^{d} \tan\{(0.5 - p_i)\pi\}/d$$

In MAAT, $d = 4$. The null distribution of *T* can be well approximated by a Cauchy distribution under arbitrary dependency structures for $p_i$, thereby the combined *p* value is calculated as $0.5 - \arctan(T)/\pi$.

### Simulation study design

We conducted comprehensive simulation analyses to compare the performance of MAAT with other state-of-the-art TWAS methods in terms of imputation $R^2$, association test power, and type I error. In the ROS/MAP dataset, we randomly selected 500 samples for the training set and 125 samples for the testing set, maintaining a 4:1 ratio between them. The genotyped and imputed genetic data for the 1000 cis-SNPs (with minor allele frequency (MAF) $\geq$ 5% and Hardy-Weinberg *p* value $\leq 1 \times 10^{-7}$) of the randomly selected gene TPTE were used as the genotype information for simulation. The gene expression profile under different settings is generated from these cis-SNPs.

We varied the proportion of causal SNPs $p_{cs}$ (0.01, 0.05, 0.1, 0.2) and the expression heritability $h_e^2$ (0.1, 0.2, 0.5) to simulate gene expression profiles, where the expression heritability refers to the proportion of gene expression variance explained by causal cis-SNPs. To assess the association test power of different methods in various scenarios, we

also changed the phenotype heritability $h_p^2$ (0.1, 0.25, 0.5), where the phenotype heritability is the proportion of phenotypic variance explained by gene expression.

For each cis-SNP, we assigned five annotation scores, two of which are informative, and three are non-informative annotations. For non-informative annotations, the annotation scores for all cis-SNPs are sampled from the same normal distribution. For informative annotations, we divided causal SNPs and non-causal SNPs into two groups. Annotation scores for cis-SNPs within the same group are sampled from the same normal distribution, while the means of the normal distributions for different groups are different. Additional file 1: Fig. S40 provides a demonstration of the simulated annotation matrix.

Gene expression and phenotype profiles are simulated based on $E_g = X\beta + \epsilon$ and $Y = \omega E_g + \epsilon_1$, respectively. Here, $X$ represents the genotype matrix, $E_g$ denotes gene expression levels for gene $g$, $Y$ is the phenotype vector, $\beta$ is the effect size vector imposed by cis-SNPs on expression, $\omega$ is the effect size imposed by GREx on phenotype, $\epsilon \sim N(0, 1 - h_e^2)$, $\epsilon_1 \sim N(0, 1 - h_p^2)$. For non-causal SNPs, their corresponding effect size values are set to 0. For causal SNPs, based on the grouping in the simulated annotation matrix, the effect sizes of the two groups of cis-SNPs are sampled from two normal distributions with mean 0 but different variances, ensuring the expression heritability to be $h_e^2$. Similarly, $\omega$ is also rescaled to achieve the target $h_p^2$.

For each ($p_{cs}$, $h_e^2$) scenario, we repeated the simulations 50 times. At each time of a specific ($p_{cs}$, $h_e^2$) scenario, where the expression level and effect size vector are fixed, we further simulated the phenotype 100 times with respect to each $h_p^2$ value. Four $p$ values were calculated with respect to each simulated phenotype based on four post-processed $\beta$ at different sparsity levels (5%, 10%, 20%, and 30%). ACAT was then employed to combine the four $p$ values into an omnibus $p$ value. The power is calculated as the proportion of $p$ values reaching the significant level among 100 replications.

To evaluate the performance of type I error, we simulated the quantitative phenotype directly from the standard normal distribution to ensure that the phenotype is independent of the genotype data. Similar to the procedure for evaluating power, for each replication in each ($p_{cs}$, $h_e^2$) scenario, the phenotype was simulated 100,000 times. The type I error rate is calculated as the proportion of $p$ values reaching the significant level among 100,000 replications.

### Real data preprocessing of ROS/MAP dataset

We applied MAAT and other TWAS methods on Religious Orders Study and Rush Memory Aging Project (ROS/MAP) dataset [27, 28]. Samples equipped with both genotype data and transcriptomic data on prefrontal cortex tissues are reserved for analysis. We used R package "bigsnpr" to perform quality control of genotype data, which enables us to execute functions provided by PLINK [98] in R. We reserved variants with minor allele frequency higher than 5% and $p$ value of Hardy-Weinberg equilibrium Fisher's exact test less than $1 \times 10^{-7}$. For a specific gene, if the proportion of samples with small FPKM (we set the threshold to be 1 in this study) accounts for more than 70%, we abandoned this gene in the TWAS analysis. We applied linear regression to adjust for gene expression levels, aiming to remove confounding effects from sex, age, study (ROS or MAP), and top three principal components (PC) of genotype data, where the top three PCs of genotype data were obtained by conducting PLINK [98]. After data

preprocessing, there were 576 samples and 10,940 genes used for the first gene expression imputation step.

### Validation in GTEx V8 database

After the imputation step in the ROS/MAP database, we tested the imputation performance for MAAT, PrediXcan, T-GEN, EpiXcan and TIGAR using the independent GTEx V8 database across 13 brain tissues. The 13 brain tissues include brain amygdala, brain anterior cingulate cortex, brain caudate basal ganglia, brain cerebellar hemisphere, brain cerebellum, brain cortex, brain frontal cortex, brain hippocampus, brain hypothalamus, brain nucleus accumbens basal ganglia, brain putamen basal ganglia, brain spinal cord cervical, and brain substantial nigra. To match the ROS/MAP database, the SNP coordinates in GTEx V8 database were converted from GRCh38/hg38 to GRCh37/hg19 reference assembly using LiftOver. To integrate the results of MAAT at different sparsity levels of $\beta$, we regarded genes with prediction $R^2$ greater than 0.005 in at least one sparsity level of $\beta$ as imputable genes.

### TF-tagged genes validation

We conducted a validation of the TF-tagged genes by integrating disease-relevant genes with the TF-gene regulatory network. Disease-relevant genes were sourced from the Phenotype-Genotype Integrator (PheGenI), and TF-gene regulatory data were obtained from TFLink (https://tflink.net/). Specifically, if a TF-tagged gene is regulated by TFs known to be associated with a specific psychiatric disease, the mechanism by which this gene affects the disease through TF regulation can be validated in some sense.

### Conservation-tagged genes validation

We sought to validate the rationality of conservation-tagged genes by referencing the housekeeping gene database. This was based on the premise that housekeeping genes exhibit conserved functions and expression patterns [67]. The set of housekeeping genes was obtained from the Housekeeping and Reference Transcript Atlas [99].

### UK Biobank data preprocessing and LD block partitioning

We adopted the methods in h2D2 [100] for preprocessing UK Biobank data and partitioning LD blocks.

### Colocalization analysis

We performed colocalization analyses using fastENLOC [33] for eight psychiatric traits. In brief, we conducted multi-SNP fine-mapping analysis using SuSiE [101] for both ROS/MAP data and GWAS summary data of eight psychiatric traits, and obtained posterior inclusion probabilities (PIPs). Then, we input eQTL PIPs and GWAS PIPs to fastENLOC and calculated gene-level colocalization probability (GLCP) for each gene-trait pair.

### Supplementary information

Additional file 1: Supplementary Materials that include additional methods, tables and figures.

Additional file 2: Table S3. List of TF-tagged genes identified by MAAT, along with the TFs regulating these genes. The TFs are well-validated to play important roles in the corresponding trait.

Additional file 3: Table S4. Full list of epigenetic-tagged genes identified by MAAT.

Additional file 4: Peer review history.

### Acknowledgements
The authors thank the editors and reviewers for their constructive comments. The Genotype-Tissue Expression (GTEx) Project was supported by the Common Fund of the Office of the Director of the National Institutes of Health, and by NCI, NHGRI, NHLBI, NIDA, NIMH, and NINDS.

### Review history
The review history is available as Additional file 4.

### Peer review information

### Authors' contributions
H.W. and Y.D.Z. conceived the project. H.W. and X.L. implemented the method and performed the analyses. H.W., X.L., T.L., Z.L., P.C.S., and Y.D.Z. interpreted the results. H.W. and Y.D.Z. drafted the first manuscript. All authors read and approved the final manuscript.

### Funding
This work was supported by Hong Kong Research Grants Council General Research Fund [17307324].

### Data availability
MAAT software is available at https://github.com/wanghanmath/MAAT under the MIT license [102]. The source code of MAAT is also deposited at Zenodo with DOI: 10.5281/zenodo.12786135 [103]. ROS/MAP data are available through Synapse with data access application (https://www.synapse.org/#!Synapse:syn3219045) [28, 104]. The protected data for the GTEx project are available via access request to the database of Genotypes and Phenotypes (accession number phs000424.v8.p2) [5, 105]. UK Biobank data are accessed under Application Number 140822 (https://www.ukbiobank.ac.uk/) [90]. The FAVOR database is available at https://favor.genohub.org [20]. The EWAS catalog database is available at http://www.ewascatalog.org/download/ [62]. The Phenotype-Genotype Integrator is at https://www.ncbi.nlm.nih.gov/gap/phegeni/. The housekeeping gene set is downloaded from https://housekeeping.unicamp.br/?homePageGlobal [99, 106]. The TF-gene regulatory data is downloaded from TFLink: https://tflink.net/ [107, 108]. The STRING protein-protein association network database is downloaded from https://string-db.org/ [52, 109].

## Declarations

### Ethics approval and consent to participate
Not applicable.

### Competing interests
The authors declare no completing interests.

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
