## [Additional file 4: Peer review history. · Genome Biology]

Review history

First round of review

Reviewer 1

In "MAAT: a new nonparametric Bayesian framework for incorporating multiple functional annotations in transcriptome-wide association studies", Wang and colleagues study the integration of different annotations to improve molecular biology imputation in GWAS and the application to TWAS, while also enabling comparison of how the mechanisms related to different annotations influence the association. They challenge other previous assumptions like positive correlations between annotation scores and SNP's effect sizes with a new quantified approach.

They apply their new method to 8 psychiatric traits and discuss their findings, along with necessary interpretation of the implicated annotations as additional biological context for novel genes implicated via MAAT. An example implementation in R is hosted here in github: <https://github.com/wanghanmath/MAAT>

The idea is valuable and the method shows promise to interrogate genetic mechanisms underlying complex traits, and can potentially be of interest to a broad audience of biologists. Unfortunately, given underlying problems common to established association techniques like the one used in this paper, additional validation of the method is needed. I'll list my concerns below explaining what I consider necessary for this submission to be considered for publication in Genome Biology.

I'll focus my review on statistical topics where I can confidently provide sound feedback. Unfortunately I lack the biological expertise necessary to consider the specific findings for psychiatric traits. The sections that discriminate associations based on the annotation (TFF, epigenetics, conservation-related, etc) are convincing to me on the value of incorporating multiple types of annotations into molecular biology prediction models.

Thank you for your attention. Sincerely,

Alvaro Barbeira

Major Concern 1: Insufficient comparison to other established methods like EpiXcan

The new method, MAAT, references other competing methods like EpiXcan ("Integrative transcriptome imputation reveals tissue-specific shared biological mechanisms mediating susceptibility to complex traits," Zhang et al 2019, reference [15] in the manuscript), which also incorporate epigenetic annotations as priors to improve prediction performance over the original generalized linear models from PrediXcan or FUSION.

1-a) I strongly recommend the authors to add EpiXcan models when doing comparisons. I.e. in the simulation section (starting line 139 and supplementary methods) that display PrediXcan, TIGAR and T-GEN; there, EpiXcan prediction models should also be considered.

1-b) To validate the increased prediction power for expression models, the authors evaluated prediction performance in the GTEx v8 database . I strongly recommend the authors add Epixcan to this comparison, and additionally, display the distribution of R^2 for models from each method

in the relevant parts of the paper and supplement, e.g. a new figure for R^2 after figure S4. (Example: Figure 3 on section 3.2 of "Fine-mapping and QTL tissue-sharing information improves the reliability of causal gene identification", Barbeira et al, Genetic Epidemiology, 2020)

Major Concern 2: TWAS vulnerabilities

The MAAT method uses an established association mechanism (from FUSION: line 127, 202, 487) for inferred molecular mechanisms on summary statistics, which like most statistical methods, is vulnerable to a complex landscape of confounders and inference problems.

One significant vulnerability is LD confounding (see "The GTEx Consortium atlas of genetic regulatory effects across human tissues", The GTEx consortium, Science 2020), when distinct causal variants for expression and traits are in LD.

Another significant vulnerability is LD mismatch when predicting associations via summary statistics, since the GWAS study's samples might have a very different LD structure to the samples used for building models, which can lead to over- or underestimating predictions (see "Opportunities and challenges for transcriptome-wide association studies" by Wainberg et al, Nature Genetics 2019, ref [4] in this manuscript). Wang and colleagues' simulations do not adequately reflect this complex scenario, so although simulations are a necessary conceptual validation, they are not sufficient.

Therefore I strongly recommend adding the following analyses to address these vulnerabilities general to all TWAS:

2-a) I find the estimation of type 1 error in simulations insufficient. I recommend adding Receiver operating characteristic analysis using a separate database of validated associations like "Online Mendelian Inheritance in Man" (see section 2.9 of Barbeira et al Genetic Epidemiology 2019 for an example)

2-b) To address LD confounding, the established practice is to complement gene-trait association methods with a separate colocalization method like COLOC, ENLOC or FOCUS (The GTEx Consortium, Science 2020). I strongly recommend incorporating colocalization results into the analysis of the section "MAAT improves the expression imputation performance [...]", in particular verifying if the associations uniquely detected by MAAT exhibit similar colocalization profiles as the other methods.

Additionally, I strongly recommend listing the colocalization results and association p-values of the novel putative genes reported in this manuscript (e.g. Table 1 and Table 2), or potentially a new figure.

Minor Concern 3: language like "Improved TWAS Power"

This manuscript focuses on a new technique for generating prediction models of molecular biology mechanisms, so I consider claims like "improved TWAS power" misleading because, as the authors claim, they use established TWAS association steps from FUSION (line 127, 202, 487). At the risk of being reiterative, "LD confounding" and "LD mismatch" between expression panels and the GWAS samples, means having more associations doesn't mean an improvement per se (as explained above in major concern 2). I would rather the authors explicitly say "Increased TWAS power to implicate associations" in instances (line 92 greater [...] TWAS power". To further delineate my concern, I give one positive example in Line 181, section title "MAAT improves the

expression imputation and identifies mode gene-trait associations" which is very clear as to the result.

Additional colocalization evidence (e.g. from ENLOC or FOCUS) could support stronger claims.

Minor concern 4: significance threshold

The authors mention a significance threshold of 5×10^{-6} in line 204. I recommend they explicitly define the threshold (e.g. "Bonferroni correction for X number of models")

Reviewer 2

The authors propose a novel TWAS tool that can integrate annotation information to improve gene expression imputation accuracy and TWAS power. The authors conducted simulation studies to show the advantages of the proposed MAAT TWAS method, compared to existing PrediXcan, TIGAR, and T-GEN method. The authors compared the proposed MAAT TWAS method with existing PrediXcan and T-GEN method. Real TWAS of eight psychiatric traits resulted in the most significant genes by MAAT, compared with PrediXcan and T-GEN. The authors also showed that the MAAT method could identify the type of annotation that are of most importance to the TWAS significant risk genes. The proposed method indeed fill in an important gap of leveraging variant annotation in TWAS. The following comments would help improve the readability and rigor of the paper:

1. The acronym PPMs shows up in the Abstract without definition, which is confusing.
2. Including the Bayesian model of the MAAT method and brief details about how annotation scores are integrated by the Bayesian model in the Results/Method overview section would help readers understand better about the method. Again, PPMx is not defined.
3. What are the genotype data used in the simulation studies should be described in the main text, including the number of SNPs. Also, the number of iterative simulations conducted for power and Type I error evaluation should be mentioned in the main text. At least 100 simulations for the power evaluation, and 10^6 NULL simulations for Type I error evaluation would be needed.
4. ROS/MAP gene expression data were profiled from the dorsolateral prefrontal cortex brain region. I do not understand why the authors would need to validate the gene expression across 13 brain tissues in the GTEx V8 data, instead of only considering the prefrontal cortex tissue?
5. In the validation and real TWAS of eight traits, why the authors not comparing with TIGAR tool?
6. What are the GWAS summary data that were used for conducting TWAS of these eight traits? Such information would be needed in the Methods section.
7. The STRING protein-protein interaction tool could help interpreting the identified TWAS risk genes of each trait, or across all eight traits. Current discussion of hundreds of (significant with $FDR < 0.05$?) enrichment pathways is blurry without enough biological evidences. It is concerning if all of these described pathways are significant with $FDR < 0.05$.
8. Manhattan plots and QQ plots with the TWAS p-values would help present the results.
9. The authors mentioned about the cost of computation by MAAT in the discussion section. Some example computation cost, including CPU time, memory usage, average computation time

per gene/chromosome would be needed.

10. It is shown that the FUSION/TWAS Z-score statistic is likely to cause inflated false positives if the eQTL weights are derived from non-standardized gene expression and genotype data (see statistic derivation in <https://www.sciencedirect.com/science/article/pii/S266624772100049X>). If so, the S-PrediXcan/TWAS Z-score statistic should be used to avoid such inflation. Please confirm which test statistic should be used with eQTL weights trained by MAAT.

Authors' response to reviewers

We appreciate the time and effort the editor and reviewers dedicated to providing feedback on our manuscript and are grateful for the insightful comments on and valuable improvements to our paper. We have incorporated/addressed all the suggestions/comments. Please see below for a point-by-point response to the comments and concerns. The comments are in black, and the responses are in blue.

Response to Reviewer #1

In "MAAT: a new nonparametric Bayesian framework for incorporating multiple functional annotations in transcriptome-wide association studies", Wang and colleagues study the integration of different annotations to improve molecular biology imputation in GWAS and the application to TWAS, while also enabling comparison of how the mechanisms related to different annotations influence the association. They challenge other previous assumptions like positive correlations between annotation scores and SNP's effect sizes with a new quantified approach.

They apply their new method to 8 psychiatric traits and discuss their findings, along with necessary interpretation of the implicated annotations as additional biological context for novel genes implicated via MAAT. An example implementation in R is hosted here in github: <https://github.com/wanghanmath/MAAT>

The idea is valuable and the method shows promise to interrogate genetic mechanisms underlying complex traits, and can potentially be of interest to a broad audience of biologists. Unfortunately, given underlying problems common to established association techniques like the one used in this paper, additional validation of the method is needed. I'll list my concerns below explaining what I consider necessary for this submission to be considered for publication in Genome Biology.

I'll focus my review on statistical topics where I can confidently provide sound feedback. Unfortunately I lack the biological expertise necessary to consider the specific findings for psychiatric traits. The sections that discriminate associations based on the annotation (TFF, epigenetics, conservation-related, etc) are convincing to me on the value of incorporating multiple types of annotations into molecular biology prediction models.

Response: Thank you so much for your thorough review and the positive feedback on our manuscript. We greatly appreciate the constructive suggestions you have provided. Your insights and suggestions have indeed helped us improve the quality and clarity of our work. We have carefully addressed your suggestions and have made the necessary revisions to address your concerns.

Major Concern 1: Insufficient comparison to other established methods like EpiXcan

The new method, MAAT, references other competing methods like EpiXcan ("Integrative transcriptome imputation reveals tissue-specific shared biological mechanisms mediating susceptibility to complex traits," Zhang et al 2019, reference [15] in the manuscript), which also incorporate epigenetic annotations as priors to improve prediction performance over the original generalized linear models from PrediXcan or FUSION.

1-a) I strongly recommend the authors to add EpiXcan models when doing comparisons.

I.e. in the simulation section (starting line 139 and supplementary methods) that display PrediXcan, TIGAR and T-GEN; there, EpiXcan prediction models should also be considered.

Response: Thanks for raising the insightful suggestion. We have added EpiXcan model in the simulation study. As shown in Fig. 2, 3 and Fig. S2, S3, EpiXcan outperforms PrediXcan in most simulation scenarios, but MAAT has better \diamond^2 and power performance in general. The newly added EpiXcan simulation results have been included in the Results/Simulation section and highlighted in red text.

1-b) To validate the increased prediction power for expression models, the authors evaluated prediction performance in the GTEx v8 database. I strongly recommend the authors add EpiXcan to this comparison, and additionally, display the distribution of R^2 for models from each method in the relevant parts of the paper and supplement, e.g. a new figure for R^2 after figure S4. (Example: Figure 3 on section 3.2 of "Fine-mapping and QTL tissue-sharing information improves the reliability of causal gene identification", Barbeira et al, Genetic Epidemiology, 2020)

Response: Thanks for the valuable suggestion. We conducted EpiXcan analysis on real data, which involved gene expression imputation using the EpiXcan model in the ROS/MAP dataset and performed R^2 validation in 13 brain tissues from the GTEx v8 dataset. The EpiXcan imputation model from the first step was subsequently applied to eight summary-level GWAS data to find significant genes relevant to eight psychiatric traits. The performance of validation \diamond^2 is shown in Fig. S4 and Fig. S5, which illustrate the number of imputable genes and the distribution of \diamond^2 across five methods respectively. We have included a detailed description of the updated real data analysis in Section "MAAT improves the expression imputation performance and identifies more gene-trait associations".

Major Concern 2: TWAS vulnerabilities

The MAAT method uses an established association mechanism (from FUSION: line 127, 202, 487) for inferred molecular mechanisms on summary statistics, which like most statistical methods, is vulnerable to a complex landscape of confounders and inference problems.

One significant vulnerability is LD confounding (see "The GTEx Consortium atlas of genetic regulatory effects across human tissues", The GTEx consortium, Science 2020) , when distinct causal variants for expression and traits are in LD. Another significant vulnerability is LD mismatch when predicting associations via summary statistics, since the GWAS study's samples might have a very different LD structure to the samples used for building models, which can lead to over- or underestimating predictions (see "Opportunities and challenges for transcriptome-wide association studies" by Wainberg et al, Nature Genetics 2019, ref [4] in this manuscript). Wang and colleagues' simulations do not adequately reflect this complex scenario, so although simulations are a necessary conceptual validation, they are not sufficient. Therefore I strongly recommend adding the following analyses to address these vulnerabilities general to all TWAS:

2-a) I find the estimation of type 1 error in simulations insufficient. I recommend adding Receiver operating characteristic analysis using a separate database of validated associations like "Online Mendelian Inheritance in Man" (see section 2.9 of Barbeira et al Genetic Epidemiology 2019 for an example)

Response: We thank the reviewer for the valuable suggestions. We have conducted the receiver operating characteristic analysis by combining Online Mendelian Inheritance in Man (OMIM) database and NHGRI-EBI GWAS catalog database [1] as silver standard. As shown in Fig. S6, except for Alzheimer's disease and anorexia nervosa, MAAT consistently ranks among the top two methods in terms of AUC across the remaining six traits evaluated. However, all methods exhibit relatively low AUC values across different traits.

On the other hand, based on our previous enrichment analysis (integrating the KEGG pathway, OMIM, KEGG disease, NHGRI GWAS catalog database, and GO database), we found that the most significant pathways for MAAT significant genes in bipolar disorder, Parkinson's disease, and schizophrenia were those related to bipolar disorder, Parkinson's disease, and schizophrenia in the NHGRI GWAS catalog database. If the false discovery rate is large, the statistical hypothesis testing method adopted in the enrichment analysis would not grant the most significant p-values for trait-related pathways.

In the section "MAAT improves the expression imputation performance and identifies more gene-trait associations", we have detailed the results of the AUC analysis and highlighted them in red.

2-b) To address LD confounding, the established practice is to complement gene-trait association methods with a separate colocalization method like COLOC, ENLOC or FOCUS (The GTEx Consortium, Science 2020). I strongly recommend incorporating colocalization results into the analysis of the section "MAAT improves the expression imputation performance [...]", in particular verifying if the associations uniquely detected by MAAT exhibit similar colocalization profiles as the other methods.

Additionally, I strongly recommend listing the colocalization results and association p-values of the novel putative genes reported in this manuscript (e.g. Table 1 and Table 2), or potentially a new figure.

Response: Thank you so much for your constructive comment. We performed colocalization analyses using fastENLOC (<https://github.com/xqwen/fastenloc>) for eight psychiatric traits. In brief, we performed multi-SNP fine-mapping analysis using SuSiE for both ROS/MAP data and GWAS summary data of eight psychiatric traits, and obtained posterior inclusion probabilities (PIPs). Then, we input eQTL PIPs and GWAS PIPs to fastENLOC and calculated gene-level colocalization probability (GLCP) for each gene- trait pair.

The distribution of GLCPs of significant genes identified by TWAS is shown in Figure R1. It turns out that most TWAS significant genes have low GLCPs. This is attributed to the limited sample size of ROS/MAP data (N=576). For most genes, SuSiE cannot detect any significant signal and the PIPs of all SNPs are low.

Figure R1, Distribution of GLCP of significant genes identified by five TWAS methods. Wilcoxon rank-sum test is performed to test the equivalence between the GLCP distribution of MAAT significant genes and other methods' significant genes.

For each trait, the number of genes with GLCP larger than 0.1 or 0.2 is shown in Table R1. We performed pathway enrichment analysis for genes with GLCP larger than 0.1 in each trait. As shown in Figure R2-R9, these genes do not show an evident enrichment in pathways related to the corresponding psychiatric disease.

Trait	Number of genes with GLCP larger than 0.1	Number of genes with GLCP larger than 0.2
Alzheimer's disease	634	1
anorexia nervosa	74	0
bipolar disorder	55	0
depression	70	0
insomnia	22	0
intelligence	52	1
Parkinson's disease	54	0
schizophrenia	43	1

Table R1, the number of genes with GLCP larger than 0.1 or 0.2

Figure R2, enriched pathways in gene set with GLCP larger than 0.1 in Alzheimer's disease

Figure R3, enriched pathways in gene set with GLCP larger than 0.1 in anorexia nervosa.

Figure R4, enriched pathways in gene set with GLCP larger than 0.1 in bipolar disorder.

Figure R5, enriched pathways in gene set with GLCP larger than 0.1 in depression.

Figure R6, enriched pathways in gene set with GLCP larger than 0.1 in insomnia.

Figure R7, enriched pathways in gene set with GLCP larger than 0.1 in intelligence.

Figure R8, enriched pathways in gene set with GLCP larger than 0.1 in Parkinson's disease.

Figure R9, enriched pathways in gene set with GLCP larger than 0.1 in schizophrenia.

The inconsistency between TWAS and colocalization analysis have been studied in this work [2] ([https://www.cell.com/ajhg/fulltext/S0002-9297\(22\)00153-7](https://www.cell.com/ajhg/fulltext/S0002-9297(22)00153-7)). Our results suggest that for the eight psychiatric traits we investigated, the inconsistency cannot be reconciled, while TWAS identified more biologically relevant genes. Therefore, we didn't include the results of colocalization in our work.

The LD confounding is indeed an important problem. However, existing TWAS fine-mapping method, like FOCUS, is not applicable to our method, because the final p-value calculated by MAAT is obtained by integrating the p-values from the effect sizes at various sparsity levels using the ACAT method, which does not match the input files for FOCUS.

Minor Concern 3: language like "Improved TWAS Power"

This manuscript focuses on a new technique for generating prediction models of molecular biology mechanisms, so I consider claims like "improved TWAS power" misleading because, as the authors claim, they use established TWAS association steps from FUSION (line 127, 202, 487). At the risk of being reiterative, "LD confounding" and "LD mismatch" between expression panels and the GWAS samples, means having more associations doesn't mean an improvement per se (as explained above in major concern 2). I would rather the authors explicitly say "Increased TWAS power to implicate associations" in instances (line 92 greater [...] TWAS power". To further delineate my concern, I give one positive example in Line 181, section title "MAAT improves the expression imputation and identifies mode gene-trait associations" which is very clear as to the result. Additional colocalization evidence (e.g. from ENLOC or FOCUS) could support stronger claims.

Response: Thanks for the valuable suggestions. We agree with the reviewer's viewpoint that using established association steps from FUSION does not imply improvements in the association step of MAAT. Therefore, we have revised the descriptions related to "improved TWAS power" to "increased TWAS power to implicate associations" as suggested by the reviewer. These changes have been made on lines 92, 460 and 506, and are highlighted in red.

Minor concern 4: significance threshold

The authors mention a significance threshold of 5×10^{-6} in line 204. I recommend they explicitly define the threshold (e.g. "Bonferroni correction for X number of models")

Response: Thanks for the kind suggestions. We redefined the p-value threshold. Since the Bonferroni-corrected significance threshold $4.57 \times 10^{-6} = 0.05/10940$ based on 10,940 genes is very close to 5×10^{-6} , we adopted 5×10^{-6} as the p-value threshold for TWAS significant genes. In line 237 of the main text, we have provided an explanation for this, which is highlighted in red.

Response to Reviewer #2

The authors propose a novel TWAS tool that can integrate annotation information to improve gene expression imputation accuracy and TWAS power. The authors conducted simulation studies to show the advantages of the proposed MAAT TWAS method, compared to existing PrediXcan, TIGAR, and T-GEN method. The authors compared the proposed MAAT TWAS method with existing PrediXcan and T-GEN method. Real TWAS of eight psychiatric traits resulted in the most significant genes by MAAT, compared with PrediXcan and T-GEN. The authors also showed that the MAAT method could identify the type of annotation that are of most importance to the TWAS significant risk genes. The proposed method indeed fill in an important gap of leveraging variant annotation in TWAS. The following comments would help improve the readability and rigor of the paper.

Response: We sincerely appreciate your detailed review and the positive feedback about our manuscript. We are grateful to the constructive comments you have provided. We have carefully addressed each of your comments and made the necessary revisions to improve the clarity and quality of our manuscript.

1. The acronym PPMs shows up in the Abstract without definition, which is confusing.

Response: Thanks for the question. PPMx stands for product partition model with covariates. We have added the full name for PPMx in the abstract section.

2. Including the Bayesian model of the MAAT method and brief details about how annotation scores are integrated by the Bayesian model in the Results/Method overview section would help readers understand better about the method. Again, PPMx is not defined.

Response: We thank the reviewer's kind suggestions. We have included Bayesian model of MAAT in the Results/Method overview section, and elaborated on how to utilize PPMx prior to incorporate annotation information into the first imputation step. Since the full name of PPMx is defined on line 77 in the Background section, we use the acronym in the following sections.

3. What are the genotype data used in the simulation studies should be described in the main text, including the number of SNPs. Also, the number of iterative simulations conducted for power and Type I error evaluation should be mentioned in the main text. At least 100 simulations for the power evaluation, and 10^6 NULL simulations for Type I error evaluation would be needed.

Response: Thanks for the great suggestions. We have included the details of the genotype data used in the simulation study in the Simulation section (line 160 to line 162). We have also listed the number of replications for evaluating the performance of R^2 , power and type I error in the main text (line 164 to line 171).

Due to time constraints, we did not have enough time to conduct 10^6 replications to evaluate the performance of type I error. We increased the number of replications from 100 before to 10^5 now. The updated simulation results are shown in Figure 2, 3 and Supplementary Figure 2, 3.

4. ROS/MAP gene expression data were profiled from the dorsolateral prefrontal cortex brain region. I do not understand why the authors would need to validate the gene expression across 13 brain tissues in the GTEx V8 data, instead of only considering the prefrontal cortex tissue?

Response: Thanks for the question. As displayed in the GTEx website

(<https://gtexportal.org/home/samplingSitePage>), there are 13 brain-related tissues, including the brain frontal cortex region, but the brain prefrontal cortex region corresponding to the ROS/MAP data is not included. Therefore, we performed analysis and validation on all 13 brain-related tissues from GTEx.

5. In the validation and real TWAS of eight traits, why the authors not comparing with TIGAR tool?

Response: Thanks for the question. We have added the TIGAR analysis in the \diamond^2 validation and real data TWAS analysis of eight traits. We have described the updated results with TIGAR and EpiXcan included in Section "MAAT improves the expression imputation performance and identifies more gene-trait associations".

6. What are the GWAS summary data that were used for conducting TWAS of these eight traits? Such information would be needed in the Methods section.

Response: Thanks for the kind suggestions. We have included the information of the GWAS summary data for eight psychiatric traits in Table S4, which provides information on the number of variants, the number of case and control samples, and the corresponding references. We also mentioned the newly added Table S4 in line 234 of the main text.

7. The STRING protein-protein interaction tool could help interpreting the identified TWAS risk genes of each trait, or across all eight traits. Current discussion of hundreds of (significant with $FDR < 0.05$?) enrichment pathways is blurry without enough biological evidences. It is concerning if all of these described pathways are significant with $FDR < 0.05$.

Response: Thanks for the great suggestions. We utilized the STRING protein-protein interaction network database and explored the interactions among the significant TWAS genes identified by MAAT. As displayed in Fig. S7, we found numerous strong interactions among the significant genes associated with different psychiatric traits. The detailed analysis is provided in the main text under the "Results/Shared significant genes in multiple traits" section, which are highlighted in red.

With respect to the pathway enrichment analysis, only pathways with their adjusted enrichment p-values less than 0.05 are displayed. Since the p-values have been adjusted, the FDR can be controlled.

8. Manhattan plots and QQ plots with the TWAS p-values would help present the results.

Response: We thank the reviewer's kind suggestions. We have included the Manhattan plots and QQ plots in the supplementary note, as shown in Figures S15-S30.

9. The authors mentioned about the cost of computation by MAAT in the discussion section. Some example computation cost, including CPU time, memory usage, average computation time per gene/chromosome would be needed.

Response: Thanks for the valuable suggestions. Since the imputation step is achieved by the Markov Chain Monte Carlo (MCMC), for a gene with approximately 3000 cis-SNPs, training the imputation model requires about 2 CPU hours on one core, similar to the computational cost of FUSION using Bayesian sparse linear mixed model (BSLMM). We have added the information in line 476 of the main text.

10. It is shown that the FUSION/TWAS Z-score statistic is likely to cause inflated false positives if the eQTL weights are derived from non-standardized gene expression and genotype data (see statistic derivation in (<https://www.sciencedirect.com/science/article/pii/S266624772100049X>). If so, the S-PrediXcan/TWAS Z-score statistic should be used to avoid such inflation. Please confirm which test statistic should be used with eQTL weights trained by MAAT.

Response: Thanks for the question. We standardized both genotype data and expression data in the first imputation step. According to the Text S2.2 section of the TIGAR-V2 paper [3], when imputation is performed using standardized data, the test statistics in S-PrediXcan and FUSION are equivalent. Therefore, we do not change the FUSION test statistics.

References

1. Buniello, A. et al. The NHGRI-EBI GWAS Catalog of published genome-wide association studies, targeted arrays and summary statistics 2019. *Nucleic acids research* 47, D1005-D1012 (2019).
2. Hukku, A., Sampson, M.G., Luca, F., Pique-Regi, R. & Wen, X. Analyzing and reconciling colocalization and transcriptome-wide association studies from the perspective of inferential reproducibility. *The American Journal of Human Genetics* 109, 825-837 (2022).

3. **Parrish, R.L., Gibson, G.C., Epstein, M.P. & Yang, J. TIGAR-V2: Efficient TWAS tool with nonparametric Bayesian eQTL weights of 49 tissue types from GTEx V8. Human Genetics and Genomics Advances 3(2022).**

Second round of review

Reviewer 1

Major concern 1-a)

I find the addition of explicit comparison to EpiXcan satisfactory. It provides valuable context when comparing methods to impute gene expression, and supports the authors' argument concerning incorporation of biologically-informed annotations into expression inference.

Major concern 1-b)

I find the addition of external validation in GTEx v8 data satisfactory, and supports the argument for incorporating biological information through annotations into expression imputation.

Major concern 2-a)

Given the nature of the OMIM and NHGRI-EBI GWAS catalogs, and the different heritability profiles for psychiatric traits, the low AUC is reasonable. I find the addition satisfactory validation of the authors' points.

Major concern 2-b)

I do not find the authors' response satisfactory.

First, it must be considered that the fastEnloc method is extremely conservative ("The GTEx Consortium atlas of genetic regulatory effects across human tissues", Science 2020). It is expected that the number of genes with $GLCP > 0.1$ to be low in most traits ("The GTEx Consortium atlas of genetic regulatory effects across human tissues", Science 2020, also evaluated this for psychiatric traits). Second, in the language of Hukku et al, AJHG 2022, the set of overlapping findings (between TWAS implications at Bonferroni correction, and, say, fastEnloc at $GLCP > 0.1$) provides a set of high confidence, biologically relevant candidate genes dubbed "conceptual replications". Merely describing this set is informative since it is considered robust against LD confounding and LD mismatch.

In the language of Hukku et al, disagreements between TWAS and colocalization don't justify discarding either method, as this response intends to do with colocalization. Hukku et al posit that differences are informative (i.e. horizontal pleiotropy). I don't think a pathway enrichment analysis preempts colocalization analysis - they can merely capture different information. One potential difference is tissue ethiology (e.g. different regions of the brain).

I recommend the authors include a brief analysis of the set of conceptual replications, such as total number of conceptual replications between enloc and different methods (EpiXcan, TIGAR, MAAT, etc). The Enriched pathways in genes with $GLCP > 0.1$ might optionally be a supplementary figure illustrating a complementary nature between TWAS

and colocalization.

I'd like to add a separate clarification, independent to the above: I don't think analyzing FOCUS fine mapping is necessary if any colocalization method is included. Using focus directly with MAAT as I understand from the authors' response is not what I intended. Instead, ROS/MAP data could be used in regular FUSION with FOCUS to do conceptual replication - however enloc itself is more than enough for this manuscript.

Minor concern 3)

I consider the response satisfactory.

Minor concern 4)

I consider the response satisfactory.

Reviewer 2

All of my comments are well addressed.

Authors' response to reviewers

We appreciate the time and effort the editor and reviewers dedicated to providing feedback on our manuscript and are grateful for the insightful comments on and valuable improvements to our paper. We have incorporated/addressed all the suggestions/comments. Please see below for a point-by-point response to the comments and concerns. The comments are in black, and the responses are in blue.

Response to Reviewer #1

I appreciate the authors' diligence in addressing my concerns. I think the additions have strengthened the manuscript and made the arguments more clear.

I consider most of my concerns addressed, but unfortunately I'm not convinced by the authors' response to "Major Concern 2-b)" below. I recommend the authors address the topics I raise in this round of review.

Response: We sincerely appreciate your detailed review and the positive feedback about our manuscript. We are grateful to the insightful suggestions you offered and have carefully addressed each of your comments. In response, we have made revisions to enhance the clarity and overall quality of the manuscript.

Major concern 1-a)

I find the addition of explicit comparison to EpiXcan satisfactory. It provides valuable context when comparing methods to impute gene expression, and supports the authors' argument concerning incorporation of biologically-informed annotations into expression inference.

Response: Thank you for your positive feedback on our comparison with EpiXcan in the simulation study.

Major concern 1-b)

I find the addition of external validation in GTEx v8 data satisfactory, and supports the argument for incorporating biological information through annotations into expression imputation.

Response: We sincerely appreciate your positive feedback on our real data analysis adding EpiXcan.

Major concern 2-a)

Given the nature of the OMIM and NHGRI-EBI GWAS catalogs, and the different heritability profiles for psychiatric traits, the low AUC is reasonable. I find the addition satisfactory validation of the authors' points.

Response: Thank you very much for your positive feedback.

Major concern 2-b)

I do not find the authors' response satisfactory.

First, it must be considered that the fastEnloc method is extremely conservative ("The GTEx Consortium atlas of genetic regulatory effects across human tissues", Science 2020). It is expected that the number of genes with GLCP > 0.1 to be low in most traits ("The GTEx Consortium atlas of genetic regulatory effects across human tissues", Science 2020, also evaluated this for psychiatric traits). Second, in the language of Hukku et al, AJHG 2022, the set of overlapping findings (between TWAS implications at Bonferroni correction, and, say, fastEnloc at GLCP > 0.1) provides a set of high confidence, biologically relevant candidate genes dubbed "conceptual replications". Merely describing this set is informative since it is considered robust against LD confounding and LD mismatch.

In the language of Hukku et al, disagreements between TWAS and colocalization don't justify discarding either method, as this response intends to do with colocalization. Hukku et al posit that differences are informative (i.e. horizontal pleiotropy). I don't think a pathway enrichment analysis preempts colocalization analysis - they can merely capture different information. One potential difference is tissue ethiology (e.g. different regions of the brain).

I recommend the authors include a brief analysis of the set of conceptual replications, such as total number of conceptual replications between enloc and different methods (EpiXcan, TIGAR, MAAT, etc). The Enriched pathways in genes with GLCP > 0.1 might optionally be a supplementary figure illustrating a complementary nature between TWAS and colocalization.

I'd like to add a separate clarification, independent to the above: I don't think analyzing FOCUS fine mapping is necessary if any colocalization method is included. Using focus directly with MAAT as I understand from the authors' response is not what I intended. Instead, ROS/MAP data could be used in regular FUSION with FOCUS to do conceptual replication - however enloc itself is more than enough for this manuscript.

Response: We thank the reviewer for the valuable suggestions. We have included a brief analysis of conceptual replications between ENLOC and five different TWAS methods. Since the number of genes with gene-level colocalization probability (GLCP) > 0.1 is very small, for each psychiatric trait, we investigated the overlap between TWAS significant genes found by five TWAS methods and the gene sets with GLCP greater than different thresholds (0.01, 0.02, ... 0.09, 0.1) across 10 levels. As shown in Figure R1 (i.e., Fig. S40), MAAT, EpiXcan, and TIGAR showed more conceptual replications. Similar to what was pointed out by Hukku et al. [1], we found that due to the presence of horizontal pleiotropy and the LD-hitchhiking effect, the number of conceptual replications is relatively low. We have added the

ENLOC results and some explanations in the Discussion section. The pathway enrichment analysis for gene sets with GLCP larger than 0.01 is shown in Supplementary Figures 41-48. We found that, after ENLOC post-processing, the enrichment levels of several key pathways increased, indicating that colocalization analysis can effectively filter out some TWAS false positives. However, we also observed that certain important pathways were lost due to very low GLCP values for some significant TWAS genes. For example, the Parkinson’s disease pathway in Parkinson’s disease was no longer detected. This highlights the complementary nature of colocalization and TWAS methods. A more detailed discussion can be found in lines 99—119 of the Supplementary Material.

Figure R1, the number of conceptual replications between ENLOC and five different TWAS methods. The x-axis represents different GLCP thresholds of ENLOC, the y-axis is the number of overlapping genes between TWAS significant genes and gene sets with GLCP greater than a specific threshold.

Minor concern 3)

I consider the response satisfactory.

Response: We sincerely appreciate your positive feedback.

Minor concern 4)

I consider the response satisfactory.

Response: We sincerely appreciate your positive feedback.

Response to Reviewer #2

All of my comments are well addressed.

Response: Thank you very much for your kind feedback.

References

1. Hukku, A., Sampson, M.G., Luca, F., Pique-Regi, R. & Wen, X. Analyzing and reconciling colocalization and transcriptome-wide association studies from the perspective of inferential reproducibility. *The American Journal of Human Genetics* 109, 825-837 (2022).

Third round of review

Reviewer 1

Concerning the item I call "Major concern 2-b)" - the spirit of the response is adequate, in the sense that there was a satisfactory analysis within the restriction to conceptual replications, as described in figure S40, and the conclusion is positive and validates the claim of MAAT's increased power within a high confidence set.

However, bringing the topic of colocalization up during the discussion was abrupt for me. I'd like to highlight the sentence in line 510: MAAT, EpiXcan and TIGAR have shown more conceptual replication compared with PrediXcan and T-GEN".

- First, "conceptual replication" is a term that appears in line 510 first without explanation so I would recommend adding a reference to Hukku et al adjacent to the new term.

- Second, at the risk of sounding reiterative, when restricting the analysis to a high-confidence set of conceptually replicated genes, MAAT performs better than PrediXcan and T-GEN, in line with EpiXcan and TIGAR. This provides additional evidence to the power of MAAT. I would consider removing this from the discussion, and instead include a simplified claim at the results section (potentially at "MAAT [...] identifies more gene-trait associations:) stating this increased power at a high confidence set of conceptual replications.

- third, the part that reads "we find that even increasing colocalization power by transitioning from variant-level to locus-level colocalization still lead to a considerable false negative rate" (line 512) is confusing to me. It is not clear to me how this was found, I merely guess it's related

to colocalization thresholds. I would recommend either removing this claim or explaining how it was found.

Authors' response to reviewers

We appreciate the time and effort the editor and reviewers dedicated to providing feedback on our manuscript and are grateful for the insightful comments on and valuable improvements to our paper. We have incorporated/addressed all the suggestions/comments in our manuscript.